# LEARNING MASK INVARIANT MUTUAL INFORMATION FOR MASKED IMAGE MODELING

**Tao Huang**[1*]  **Yanxiang Ma**[1*]     **Shan You**[2]     **Chang Xu**[1†]
[1]School of Computer Science, Faculty of Engineering, The University of Sydney
[2]SenseTime Research

## ABSTRACT

Masked autoencoders (MAEs) represent a prominent self-supervised learning paradigm in computer vision. Despite their empirical success, the underlying mechanisms of MAEs remain insufficiently understood. Recent studies have attempted to elucidate the functioning of MAEs through contrastive learning and feature representation analysis, yet these approaches often provide only implicit insights. In this paper, we propose a new perspective for understanding MAEs by leveraging the information bottleneck principle in information theory. Our theoretical analyses reveal that optimizing the latent features to balance relevant and irrelevant information is key to improving MAE performance. Building upon our proofs, we introduce MI-MAE, a novel method that optimizes MAEs through mutual information maximization and minimization. By enhancing latent features to retain maximal relevant information between them and the output, and minimizing irrelevant information between them and the input, our approach achieves better performance. Extensive experiments on standard benchmarks show that MI-MAE significantly outperforms MAE models in tasks such as image classification, object detection, and semantic segmentation. Our findings validate the theoretical framework and highlight the practical advantages of applying the information bottleneck principle to MAEs, offering deeper insights for developing more powerful self-supervised learning models.

## 1 INTRODUCTION

Masked autoencoders (MAEs) (He et al., 2022; Xie et al., 2022; Bao et al., 2022) have emerged as a powerful self-supervised learning paradigm, particularly in the realm of computer vision. Inspired by the success of masked language models like BERT (Devlin et al., 2019) in natural language processing, MAEs leverage a similar masking and reconstruction strategy to learn meaningful visual representations. The fundamental concept involves masking a portion of the input image and training a model to predict the missing parts, thereby enabling the model to capture the underlying structure and semantics of the visual data. This approach has proven effective in numerous applications (Li et al., 2022b; Kirillov et al., 2023; Tong et al., 2022; Fang et al., 2023b), showcasing the potential of MAEs to learn robust and generalizable features from unlabeled data.

Despite their success, the understanding of how MAEs function and why they perform well remains an open question. Recent research has sought to demystify the inner workings of MAEs, providing valuable insights into their operation. Several studies have approached this task from various perspectives, including contrastive learning (Zhang et al., 2022; Kong & Zhang, 2023; Huang et al., 2023) and feature representation analysis (Xie et al., 2023; Pan et al., 2023). Specifically, (Kong & Zhang, 2023) proposed that MAEs inherently learn occlusion-invariant features by treating masked patches as a form of data augmentation. This approach, also suggested by (Zhang et al., 2022; Yue et al., 2023), aligns MAEs with contrastive learning frameworks, where the models learn to align features between different masked views of the same image. Other studies such as (Xie et al., 2023; Pan et al., 2023) analysed the latent feature representations learned by MAEs to understand how these models capture and organize visual information. However, these efforts often provide only

---

[*]Equal contributions. [†]Corresponding author.

implicit insights and do not fully address the need for a comprehensive and systematic understanding of the learning objectives and framework of MAEs.

In this paper, we propose a new perspective for understanding MAEs by leveraging information theory, specifically the information bottleneck (IB) principle (Tishby & Zaslavsky, 2015). This perspective provides a systematic and comprehensive framework for optimizing MAEs, offering theoretical insights that can guide the development of more effective models. The IB principle posits that any deep neural network can be understood as a system that balances the trade-off between retaining relevant information and compressing irrelevant information. By applying this principle to MAEs, we aim to provide a more robust understanding of their mechanisms and to identify key areas for improvement.

Based on our findings, we introduce a novel masked image modeling method, dubbed MI-MAE, to learn Mask Invariant Mutual Information for MAEs through the lens of the information bottleneck theory. Our method systematically optimizes the latent features produced by the encoder, ensuring that they contain maximal relevant information and minimal irrelevant information on the information bottleneck of MAE. Concretely, we introduce two aspects of mutual information based losses on the latent feature: (1) Mutual information maximization. To optimize the autoencoder in the latent space, we derive a loss to maximize the mutual information between the latent features of multiple orthogonal masks[1]. (2) Mutual information minimization. We optimize an upper bound of mutual information between the input and latent space to minimize the irrelevant information in the latent features and thus maximize the capacity of relevant information. This comprehensive optimization strategy helps in achieving better feature representations and improved performance.

We conduct a series of evaluations on standard benchmarks, showing that our method performs significant improvements over MAE in various tasks, including image classification, object detection, and semantic segmentation. For example, our 400-epoch model achieves $83.9\%$ accuracy on ImageNet-1K, surpassing the 1600-epoch MAE by $0.5\%$. The experimental results validate our theoretical findings and highlight the practical benefits of applying the information bottleneck principle to masked autoencoders. Additionally, by providing a new perspective and a rigorous analytical framework, our work paves the way for future research in this area, offering insights that can drive the development of even more powerful self-supervised learning models.

## 2 RELATED WORKS

**Contrastive learning.** Contrastive learning (Chen et al., 2020; He et al., 2020; Chen & He, 2021; Grill et al., 2020; Caron et al., 2021) stands out as the leading self-supervised representation learning approach in computer vision, achieving invariance by comparing different augmentations of the same image. A notable example is SimCLR (Chen et al., 2020), which enhances semantic representations by increasing the similarity between various views of the same image in the latent space. MoCo v3 (Chen et al., 2021) applies contrastive learning techniques to pre-train vision transformers. DINO (Caron et al., 2021) delves into novel properties of self-supervised vision transformers.

**Masked image modeling.** Masked image modeling (MIM) has gained significant traction in the field of computer vision as an effective self-supervised learning paradigm. Recently, with the widespread use of vision transformers (ViTs) (Dosovitskiy et al., 2021; Liu et al., 2021), a series of notable methods such as BEiT (Bao et al., 2022), MAE (He et al., 2022), and SimMIM (Xie et al., 2022) have been proposed to pre-train ViTs following the BERT-style masked modeling paradigm used in natural language processing (NLP) (Devlin et al., 2019; Liu et al., 2019). Many follow-up works extend masked pre-training by exploring data augmentations (Chen et al., 2023; Fang et al., 2023a), mask strategies (Li et al., 2022a; Wang et al., 2023; 2024), and hierarchical structures (Xie et al., 2022; Huang et al., 2022; Woo et al., 2023). Additionally, there is growing interest in understanding MAE and its connection with contrastive learning (Zhang et al., 2022; Xie et al., 2023; Huang et al., 2023; Kong & Zhang, 2023; Pan et al., 2023). In this paper, we further investigate MAE from an information bottleneck perspective.

**Information bottleneck.** Under information theory, any closed system can be quantified by the mutual information between bottleneck and output variables (Tishby et al., 2000). A DNN with a

---

[1]Here, "orthogonal" means the inner productions between each mask are 0, which means we are completely dividing the image into visible parts of multiple masks.

given input can be considered as a closed system that introduces no other information. During the forward propagation, the complexity of the intermediate variables usually decreases in a general prediction model, as does the amount of information they contain. It is possible to measure the goodness of each layer and even the whole prediction network by the mutual information that can be used between the intermediate variables or the outputs and the network's prediction target (Tishby & Zaslavsky, 2015).

## 3 PRELIMINARIES

Masked Autoencoders (MAEs) (He et al., 2022) are a type of self-supervised model designed to reconstruct masked patches in images. These autoencoders consist of two main components: an encoder, which encodes the image into latent features, and a decoder, which predicts the masked patches. During training, each input image is first embedded into a feature representation $X$, which is divided into multiple patches. A random mask $m$ is then generated to select a subset of these patches as visible patches. The visible patches, represented as $X \cdot (1 - m)$, are concatenated with a learnable class token and fed into a Vision Transformer (ViT) encoder to obtain the latent feature $z$.

Subsequently, the latent feature $z$ is concatenated with a set of learnable mask tokens representing the masked patches and passed into a decoder to predict the original unmasked patches $\hat{X}$. Finally, a linear projector is used to generate the reconstructed image $\phi(\hat{X})$. The loss function for training MAEs is based on the reconstruction errors between the original masked pixels and the predicted masked pixels:

$$\mathcal{L}_{\text{rec}} = ||o(X \cdot m) - \phi(\hat{X} \cdot m)||_2^2, \tag{1}$$

where $o(X \cdot m)$ denotes the original pixel values of the image associated with the masked patches $X \cdot m$.

**Approaches in understanding MAEs.** Recent studies have provided several insights into the functioning of Masked Autoencoders (MAEs), with works mainly understanding MAEs from the perspective of contrastive learning. For instance, (Kong & Zhang, 2023) proposed that MAEs inherently learn occlusion-invariant features by treating masked patches as a form of data augmentation. This perspective aligns MAEs with contrastive learning frameworks, suggesting that MAEs implicitly align features between different masked views of the same image. Similarly, U-MAE (Zhang et al., 2022) established a theoretical connection between MAEs and contrastive learning, showing that the reconstruction loss of MAEs aligns well with the alignment of mask-induced positive pairs, thereby enhancing feature uniformity and diversity. However, these papers only provide implicit analyses of MAEs by introducing additional views of the image to justify that the MAEs implicitly align with contrastive learning, and the introduced methods only perform on par with the original MAE. For a comprehensive understanding of MAEs, further analyses of the learning objective and autoencoder framework are needed.

Taking the above approaches into account, we conclude that optimizing the latent features produced by the encoder is crucial for improving MAEs. This motivates us to perform an in-depth analysis of MAEs and the latent features using information theory. In this paper, we provide a more comprehensive and theoretically sound analysis following the information bottleneck principle and show that the key to improving MAEs is maximizing relevant information while compressing irrelevant information in the latent space. Through our analysis with information theory, we find that the contrastive learning on latent space can help minimize the IB distortion

## 4 METHOD

In Section 4.1, we introduce the information theorem to explain the workings of MAEs. The information bottleneck principle, as introduced by (Tishby & Zaslavsky, 2015), is employed and customized for MAEs to elucidate the overall training objective via the information bottleneck framework. In Section 4.2, we break down the overall objective and propose two new loss functions for MAEs, based on the assumptions considering the information bottleneck within MAEs.

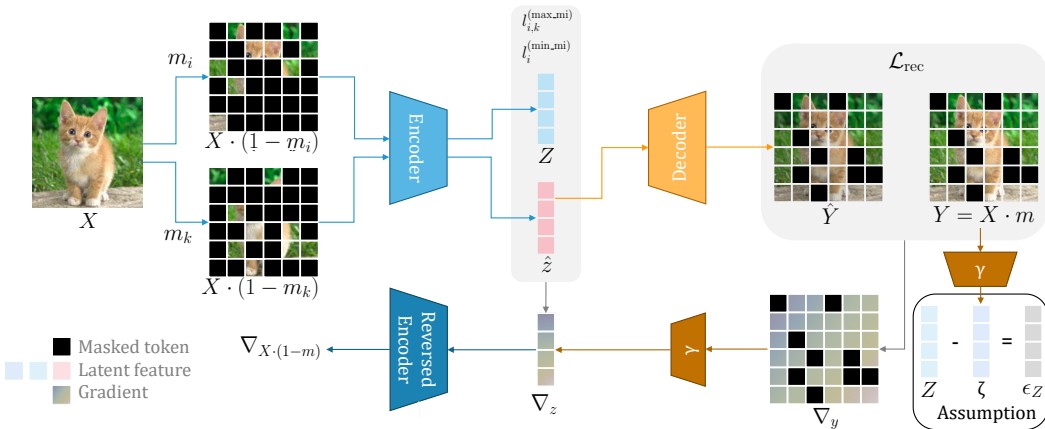

Figure 1: Pipeline of MI-MAE for each mask $m_k$. We introduces two losses $l_{i,k}^{(\text{max\_mi})}$ and $l_i^{(\text{min\_mi})}$ on the latency to maximize the relevant information and minimize the irrelevant information respectively, and $\mathcal{L}_{\text{rec}}$ denotes the original MAE loss. The top sequence in the figure denotes forward propagation and the bottom denotes back propagation. $m$ denotes mask and $X$ denotes the original map. $\gamma$ is the inverse function of a decoder, $\zeta$ is the output of the reduced target map of the MAE on $\gamma$, and $Z$ is defined as a latent feature on a small neighbourhood of $\zeta$, and their bias $\varepsilon_z$ is decided by $\epsilon_y$. $\nabla$ in backpropagation represents gradients, while $\nabla_h$ is the gradient in layer $h$ of the encoder.

## 4.1 MAE WITH INFORMATION BOTTLENECK PRINCIPLE

In information theory, deep neural networks suffer from information distortion as the information complexity of intermediate variables decreases (Tishby et al., 2000; Tishby & Zaslavsky, 2015).

**Definition 1.** *Based on the definition of notation in Section 3, the information distortion in MAEs is defined as*

$$D_I = I(X \cdot (1-m); X \cdot m | \widehat{X \cdot m}), \tag{2}$$

*where $\widehat{X \cdot m}$ is the prediction and $I(\cdot; \cdot)$ denotes the mutual information between two variables.*

The information distortion describes the portion of the mutual information between the masked image and the unmasked image that is not captured by the recovered image. For a given preset, training the neural network reduces the information distortion. The internal variable that captures all the mutual information between the masked image and the recovered image is called the effective description, with the one having the least information complexity referred to as the simplest effective description (Tishby & Zaslavsky, 2015). For any given MAE and training data, the information distortion is determined by the simplest effective distortion, denoted by $X \cdot \widetilde{(1-m)}$. The simplest effective distortion is considered as the information bottleneck (IB) in MAEs. Thus, information distortion is limited by the IB as $D_{IB} = I(X \cdot (1-m); X \cdot m | X \cdot \widetilde{(1-m)})$. According to (Tishby et al., 2000; Tishby & Zaslavsky, 2015), the goal of the MAE can be re-interpreted as minimizing a Lagrangian term that includes $D_{IB}$, formulated as:

$$L[p(\hat{x}|x)] = I(X \cdot (1-m); X \cdot \widetilde{(1-m)}) + \beta D_{IB}. \tag{3}$$

In this Lagrangian term, the first sub-term represents the complexity of the simplest effective description of the samples, and the second sub-term represents the information distortion of the given network. It is challenging to precisely find $X \cdot \widetilde{(1-m)}$ by training MAE on a given data distribution (Tishby & Zaslavsky, 2015). The MAE can only find a sub-optimal effective description in the neighborhood of $X \cdot \widetilde{(1-m)}$.

**Theorem 2.** *Denote $X \cdot \widetilde{(1-m)} + r$ as a biased simplest effective description found through training, where $r$ is the bias. Let the predicted latent feature for the MAE be $\hat{z}$. The latent feature is*

*the information bottleneck for the MAE, and thus $\hat{z} = X \cdot \widetilde{(1-m)} + r$. The mutual information $I(X \cdot \widetilde{(1-m)}; X \cdot m)$ can be upper bounded by a generalization bound as:*

$$I(X \cdot \widetilde{(1-m)}; X \cdot m) \leqslant \hat{I}(\hat{z}; X \cdot m) + O(\frac{K_x|Y|}{\sqrt{n_x}}) - I(\hat{z}; X \cdot m|r), \qquad (4)$$

*where $K_x = |\widetilde{X}|$ denotes the complexity of $X \cdot \widetilde{(1-m)}$, $n_x$ is the size of $X \cdot (1-m)$, and $\hat{I}$ is the empirical estimate of the mutual information from the given training set.*

The proof of Theorem 2 is in Appendix A.1. From the upper bound, it can be seen that mitigating the bias on the information bottleneck helps in achieving better MAE performance. Unfortunately, optimizing the latent feature encounters the problem of difficulty in finding the optimal latent feature. In the following, we will analyse how to learn using a sub-optimal latent feature.

## 4.2 MULTIPLE OBJECTIVES WITH INFORMATION THEOREM

Define an optimal simplest effective description as $\zeta = X \cdot \widetilde{(1-m)}$. Maximizing the mutual information between the latent feature and $\zeta$ will help reduce $I(\hat{z}; X \cdot m|r)$. Unfortunately, it is challenging to find a precise $\zeta$ in the latent space for all samples. One approach is to use a sub-optimal latent feature.

**Assumption 3.** *The MAE loss has been minimized on the given training set, i.e. $\mathcal{L}_{\text{rec}} \leqslant \epsilon_l$. Denote $Y$ as the ground truth of the MAE, and $\hat{Y}$ as the prediction. An upper bound of the information distortion at the output layer can be found as:*

$$H(Y|\hat{Y}) \leqslant \epsilon_Y, \qquad (5)$$

*where $H(Y)$ is the information in $Y$, $\epsilon_l$ is a small constant, and $\epsilon_Y$ is determined by $\epsilon_l$.*

Proof of the validity of Assumption 3 is shown in the Appendix. Under Assumption 3, we can find a set of sub-optimal latent features for $X \cdot (1-m)$, defined as $Z$, as the target for the samples whose latent feature is similar to $Z$. Furthermore, $|Z - \zeta| \leqslant \epsilon_z$. This idea is similar to contrastive learning. For a set of samples with similar relevant information in the ground truth, the relevant information in the information bottleneck is also similar in the same MAE. To make such a set, we consider generating multiple masks for a certain image before training and keeping the masks invariant. Consider the following case: for a given image $X$, generate $N$ mutually orthogonal masks as a pre-selected set $M = \{m_1, m_2, ..., m_N\}$. For each generated mask $m_i$, the input is denoted $X_i = X \cdot (1-m_i)$. In particular, represent $X_0$ as the part of $X$ that is not included in any input, *i.e.*, $X_0 = X - \sum_{i=1}^{n} X_i$. Then the ground truth can be expressed as $X \cdot m_i = \sum_{n; j \neq i}^{j=0} X_j$. Let each mask $m_i$ correspond to an optimal latent feature $z_i = \zeta$, whose prediction is $\hat{z}_i$. Note that the prediction of the latent feature is the biased simplest effective description of $X_i$, and the optimal latent feature is the simplest effective description of $X_i$.

**Corollary 4.** *With Assumption 3 standing, there exists a certain mask $m_i$, whose predicted latent feature $\hat{z}_i = Z$. The mutual information between the prediction and the optimal latent feature for the other masks is*

$$I(\hat{z}_k; z_k) \leqslant l_i + I(\hat{z}_k; \hat{z}_i) - I(\hat{z}_i; X_0|z_k) - \sum_{n; j \notin \{i,k\}}^{j=1} [I(\hat{z}_i; X_j|z_k) - I(\hat{z}_k; X_j|z_i)], \qquad (6)$$

*where $l_i = I(\hat{z}_i; X_0) + \sum_{n; j \notin \{i,k\}}^{j=1} I(\hat{z}_i; X_j)$. For a given $z_i$, $l_i$ is a fixed value. $I(\hat{z}_k; z_k)$ can be **maximized** only when the three following conditions are satisfied: (1). $I(\hat{z}_k; \hat{z}_i)$ is **maximized**; (2). $I(\hat{z}_i; X_0|z_k)$ is **minimized**; (3). for any $j$ that satisfies $j \in \mathbb{Z} \cap [1, N]$ and $j \notin \{i, k\}$, $\sum_{n; j \notin \{i,k\}}^{j=1} I(\hat{z}_k; X_j|z_i)$ is **maximized**, where $\mathbb{Z}$ denotes the set of integers.*

---

**Algorithm 1** Self-supervised pre-training with MI-MAE. Our changes to MAE are marked with *.

---

**Input:** Encoder $\mathbb{E}$, decoder $\mathbb{D}$, variational distribution approximation network $\mathbb{V}$ with parameters $\theta$, training dataset $\mathcal{D}_{tr}$, number of masks per image $N$.

1: **for** iteration in total_iterations **do**
2:     $X \leftarrow \mathcal{D}_{tr}$;                             *# Sample a batch of images from training set*
3:     * Generate $N$ orthogonal masks $M = \{m_1, m_2, ..., m_N\}$ for each image;
4:     Encode the masked images, $\hat{z}_i \leftarrow \mathbb{E}(X \cdot (1 - m_i)), \forall 1 \leq i \leq N$;
5:     Decode the latents, $\hat{Y} \leftarrow \mathbb{D}(\hat{z}_i)$;
6:     * Predict variational distribution $q_\theta(\hat{z}|X) \leftarrow \mathcal{N}(\hat{z}; \mu(X; \theta), \sigma(X; \theta))$;
7:     Compute $\mathcal{L}_{\text{rec}}$, $\mathcal{L}_{\text{max\_mi}}$, and $\mathcal{L}_{\text{min\_mi}}$ with $X, M, \hat{z}, q_\theta(\hat{z}|X)$;
8:     * Optimize encoder $\mathbb{E}$ with $\nabla(\lambda_1 \mathcal{L}_{\text{rec}} + \lambda_2 \mathcal{L}_{\text{max\_mi}} + \lambda_3 \mathcal{L}_{\text{min\_mi}})$ (Eq. 11);
9:     Optimize decoder $\mathbb{D}$ with $\nabla \lambda_1 \mathcal{L}_{\text{rec}}$;
10:    * Optimize approximation network $\mathbb{V}$ with $\nabla \mathcal{L}_{\text{approx}}$;
11: **end for**
**Output:** Trained encoder $\mathbb{E}$ and decoder $\mathbb{D}$.

---

From the first condition of the mutual information maximization in Corollary 4, we can adopt In-foNCE, a widely-used loss to maximize the lower bound of mutual information (Oord et al., 2018), *i.e.*,

$$l_{i,k}^{(\text{max\_mi})} = -\log \frac{\exp(\text{sim}(\hat{z}_i, \hat{z}_k)/\tau)}{\sum_{c=1}^{NB} \mathbb{1}_{[c \neq i]} \exp(\text{sim}(\hat{z}_i, \hat{z}_c)/\tau)}, \tag{7}$$

where $\text{sim}(u, v) = u^\top v/\|u\|\|v\|$ denotes the cosine similarity between two feature vectors, $\mathbb{1}[c \neq i]$ is an indicator function that evaluates to 1 if and only if $c \neq i$, $B$ denotes the batch size, $NB$ is the total number of masked images with $N$ masks per image, and $\tau$ is a temperature factor. We set $\tau = 0.07$ in all experiments. Therefore, the final MI maximization loss among all the image pairs is formulated as

$$\mathcal{L}_{\text{max\_mi}} = \frac{1}{N^2} \sum_{i=1}^{N} \sum_{k=1}^{N} \mathbb{1}_{[i \neq k]} l_{i,k}^{(\text{max\_mi})}. \tag{8}$$

Considering the first term in Eq. 3, we also need to minimize the mutual information between the latent feature and the masked image. Unlike $l_{i,k}^{(\text{max\_mi})}$, the mutual information about $\hat{z}_j$ and $X_j$ cannot be represented by the cosine similarity, since they are not in the same feature space. Therefore, for the minimization of MI, we use the Mutual Information Neural Estimator (MINE) (Belghazi et al., 2018) to represent the mutual information in KL divergence. This $I(\hat{z}_k; X_j)$ representation requires prior probability $p(\hat{z}_j|X_j)$. Since the prior probability is intractable, we follow (Kingma & Welling, 2013; Cheng et al., 2020) and use an approximation neural network to estimate the variational distribution of $p(\hat{z}_j|X_j)$, where the loss function for minimizing it is the negative log-likelihood between $z_i$ and $X_i$, *i.e.*,

$$\mathcal{L}_{\text{approx}} = \frac{1}{N} \sum_{j=1}^{N} -\log q_\theta(z_j|X_j), \tag{9}$$

where $\theta$ is the parameters in the approximation network. With the estimated posterior probability, we use the upper bound of MI presented in CLUB (Cheng et al., 2020) to minimize $I(\hat{z}_k; X_j)$, for all $j$, we aim to minimize:

$$\mathcal{L}_{\text{min\_mi}} = \frac{1}{N} \sum_{j=1}^{N} l_j^{(\text{min\_mi})} \quad \text{with } l_j^{(\text{min\_mi})} = \log q_\theta(\hat{z}_j|X_j) - \frac{1}{N} \sum_{k=1}^{N} \log q_\theta(\hat{z}_k|X_j). \tag{10}$$

Detailed derivations for the upper bound can be found in Appendix A.4. Additionally, we find that by optimizing Eq. 10, the third condition in Corollary 4 is also satisfied.

In addition, Assumption 3 needs the MAE loss to be limited to a small value. Thus, we should also add the original MAE loss as a part of the training loss. Considering Assumption 3, with both parts of the Lagrangian term minimized, our final loss becomes

$$\mathcal{L}_{\text{MI-MAE}} = \lambda_1 \mathcal{L}_{\text{rec}} + \lambda_2 \mathcal{L}_{\text{max\_mi}} + \lambda_3 \mathcal{L}_{\text{min\_mi}}, \tag{11}$$

Table 1: Results on ImageNet classification task. The backbone for SimMIM-based methods is Swin-B (Liu et al., 2021), while others are ViT-B (Dosovitskiy et al., 2021). ∗: The 800-epoch MAE results are reported by MFF (Liu et al., 2023) based on running the official code of MAE.

| Method | Epochs | ImageNet ACC (%) | | |
|---|---|---|---|---|
| | | FT | LIN | $FT_{1\%}$ |
| Supervised | - | 81.8 | - | - |
| DINO (Caron et al., 2021) | 800 | 82.8 | 78.2 | - |
| MoCo v3 (Chen et al., 2021) | 300 | 83.2 | 76.7 | 63.4 |
| BEiT (Bao et al., 2022) | 800 | 83.2 | - | - |
| C-MAE (Kong & Zhang, 2023) | 400 | 83.2 | - | - |
| SemMAE (Li et al., 2022a) | 800 | 83.3 | 65.0 | - |
| MFF (Liu et al., 2023) | 800 | 83.6 | 67.0 | 48.0 |
| MAE* (He et al., 2022) | 800 | 83.3 | 65.6 | 45.4 |
| **MI-MAE** | 200 | 83.9 (+0.6) | 67.9 (+2.3) | 48.2 (+2.8) |
| MAE (He et al., 2022) | 1600 | 83.6 | 68.0 | 51.1 |
| **MI-MAE** | 400 | 84.1 (+0.5) | 69.3 (+1.3) | 52.3 (+1.2) |
| PixMIM (Liu et al., 2024) | 800 | 83.5 | 67.2 | 47.9 |
| SimMIM (Xie et al., 2022) | 800 | 83.8 | 56.7 | - |
| **MI-SimMIM** | 400 | 84.1 (+0.3) | 59.1 (+2.4) | 49.1 (+1.2) |

where $\lambda_1$, $\lambda_2$, and $\lambda_3$ are hyper-parameters for balancing the loss terms. Considering all the losses, the pipeline of our MI-MAE is shown in Fig. 1. To be more specific, for the major MAE, we add $\mathcal{L}_{\min\_mi}$ and $\mathcal{L}_{\max\_mi}$ to the latent features after the forward propagation of the encoder. Before $\mathcal{L}_{\min\_mi}$ is calculated, the approximation network is deployed to get the posterior probabilities $q_\theta(\hat{z}_j|X_j)$ and $q_\theta(\hat{z}_k|X_j)$. Note that the gradient of $\mathcal{L}_{\min\_mi}$ and $\mathcal{L}_{\max\_mi}$ will only influence the encoder in back propagation. After that, the decoder will recover the latent feature to the image space. $\mathcal{L}_{rec}$ will then be used to influence the whole MAE in back propagation. The training process of our model is illustrated in Algorithm 1. Specifically, the size of $X_0$ in our method is set to 0 to satisfy the second condition in Corollary 4.

## 5 EXPERIMENTS

### 5.1 EXPERIMENT SETUP

To sufficiently validate the efficacy of our method, we conducted a series of experiments on image classification, object detection, and semantic segmentation tasks.

**Image classification.** Our method is developed based on the official code of MAE (He et al., 2022). We strictly adhere to the original pre-training and fine-tuning settings on ImageNet-1K (Russakovsky et al., 2015).

- **Pre-training.** Since our method samples four masks for every image, for fair comparisons, we reduce our training epochs to one-quarter of the epochs used in the compared models. Specifically, we pre-train the models using an AdamW optimizer (Loshchilov & Hutter, 2019) with $\beta_1 = 0.9$, $\beta_2 = 0.95$, and a weight decay of 0.05. The total batch size is 1024 (equivalent to 4096 as we augment each image with 4 masks inside the model). We use a cosine decay learning rate schedule with a 10-epoch warmup and a base learning rate of $1.5 \times 10^{-4}$. For the hyper-parameters introduced by our MI-MAE, we set $\lambda_1 = \lambda_2 = 1$ and $\lambda_3 = 10$. For Assumption 3, $\epsilon_l$ is set to 0.5. This means that we only use $\mathcal{L}_{rec}$ before $\mathcal{L}_{rec}$ is less than 0.5, and the entire loss is used after $\mathcal{L}_{rec}$ meets the assumption. We set $N = 4$, which means four orthogonal masks are generated in each iteration for each image with a masking ratio of 0.75. We also conduct experiments on SimMIM (Xie et al., 2022) architecture with a masking ratio of 0.5, and the $N$ is set to 2 accordingly.

- **Fine-tuning.** The base learning rate for fine-tuning is set to $1 \times 10^{-3}$. We warm up the learning rate for five epochs and train the models for a total of 100 epochs with an overall batch size of 1024. Stronger augmentations and regularization such as RandAug-

Table 2: Results on COCO instance segmentation and ADE20K semantic segmentation. The backbone of all methods is ViT-B. The results of MoCo v3 and BEiT are from MAE (He et al., 2022). The COCO results of MAE are from ViTDet (Li et al., 2022b), and our method uses the same architecture and training strategy.

| Method | Pre-train data | COCO | | ADE20K |
| | | $AP^{box}$ | $AP^{mask}$ | mIoU |
|---|---|---|---|---|
| Supervised | IN1K w/ labels | 47.9 | 42.9 | 47.4 |
| MoCo v3 | IN1K | 47.9 | 42.7 | 47.3 |
| BEiT | IN1K+DALLE | 49.8 | 44.4 | 47.1 |
| MAE | IN1K | 51.2 | 45.5 | 48.1 |
| **MI-MAE** | IN1K | **52.0** | **46.1** | **49.3** |

ment (Cubuk et al., 2020), label smoothing (Szegedy et al., 2016), and mixup (Zhang et al., 2018) are adopted.

- **Linear probing.** For linear probing, we use the pre-trained and fixed feature of the class token to learn a linear predictor. We use the LARS optimizer (You et al., 2017) with a base learning rate of $0.1$ and a batch size of $16384$. The weight decay is set to $0$. We train the linear probing for 90 epochs with a 10-epoch warmup. All our experiments use NVIDIA V100 GPUs.

**Object detection.** We transfer the pre-trained ViT models to COCO (Lin et al., 2014) dataset. We adopt Mask R-CNN framework (He et al., 2017), which predicts detections and instance segmentations simultaneously. We follow the model setup and training strategy used in ViTDet (Li et al., 2022b).

**Semantic segmentation.** We conduct semantic segmentation experiments on the ADE20K (Zhou et al., 2017) dataset, using the same settings as in MAE (He et al., 2022). Specifically, we fine-tune UperNet (Xiao et al., 2018) for 160k iterations with a batch size of 16.

## 5.2 MAJOR RESULTS

**Image classification.** We conduct pre-training on the ImageNet dataset and report the fine-tuning, linear probing, and low-shot (1% samples) fine-tuning accuracies in Tab. 1. After pre-training ViT-B for 200 epochs (equivalent to 800 epochs of MAE), our method achieves significant improvements of 0.6, 2.3, and 2.8 percentage points on fine-tuning, linear probing, and low-shot fine-tuning, respectively. Compared to the optimal 1600-epoch MAE, our method still surpasses it by 0.5, 1.3, and 1.2 percentage points, obtaining an outstanding final fine-tuning accuracy of 84.1%. We also implement our method on SimMIM (Xie et al., 2022) framework, another typical masked image modeling method with hierarchical Swin model (Liu et al., 2021). The results show that, our MI-SimMIM achieves 84.1% accuracy, outperforming previous methods such as PixMIM and SimMIM.

To evaluate the transfer learning performance of our method, we apply our pre-trained 400-epoch model to downstream tasks on the COCO and ADE20K datasets.

**Object detection and instance segmentation.** Tab. 2 reports the bounding box AP and mask AP performance on COCO detection. Compared to MAE, our method achieves significant improvements of 0.8 and 0.6 in $AP^{box}$ and $AP^{mask}$, respectively. This validates our method's superiority in dense prediction tasks.

**Semantic segmentation.** We also report the performance on the ADE20K segmentation task in Tab. 2. Our method achieves a notable improvement of 1.2 mIoU compared to MAE, demonstrating its superiority in discriminating semantic pixels.

## 5.3 ABLATION STUDY

To investigate the contributions of each innovation in our method and aid in determining the design choices, we conduct ablation experiments on our method. All experiments are performed with ViT-B on ImageNet-1K. We pre-train the models for 50 epochs, while for our MAE baseline, we train

Table 3: Ablation experiments with ViT-B on ImageNet-1K. All the models are pre-trained for 50 epochs and fine-tuned for 100 epochs. We run the original MAE for 200 epochs for comparison. Default settings are marked in  gray .

(a) Losses.

| Combination | $\mathcal{L}_{\text{rec}}$ | $l_{i,k}^{(\text{max\_mi})}$ | $l_i^{(\text{min\_mi})}$ | $\mathcal{L}_{\text{approx}}$ | ACC |
|---|---|---|---|---|---|
| (a) | ✓ | - | - | - | 82.2 |
| (b) | ✓ | ✓ | - | - | 82.5 |
| (c) | - | ✓ | - | - | Collapse |
| (d) | ✓ | ✓ | ✓ | - | 82.4 |
| (e) | ✓ | - | ✓ | ✓ | 82.4 |
| (f) | ✓ | ✓ | ✓ | ✓ | **82.8** |

(b) Loss weights.

| $\lambda_2$ | $\lambda_3$ | ACC |
|---|---|---|
| 1 | 1 | 82.6 |
| 1 | 10 | **82.8** |
| 1 | 20 | 82.7 |
| 0.1 | 10 | 82.5 |
| 0.5 | 10 | 82.7 |
| 10 | 10 | 82.6 |

(c) Mask generation.

| Type | ACC |
|---|---|
| Independent | 82.6 |
| Orthogonal | **82.8** |

the model for 200 epochs to match the number of mask samples in our methods. We compare the 100-epoch fine-tuning accuracies of the models.

**Ablation on the proposed losses.** As reported in Tab. 3 (a), we pre-train the models with different combinations of losses proposed in our method and obtain the following findings:
(1) Compared to the original MAE (a), maximizing the mutual information between the latent features of an image in (b) results in a 0.3 percentage point improvement.
(2) Using $l_{i,k}^{(\text{max\_mi})}$ only in (c) results in a collapse of training, *i.e.*, the autoencoder cannot reconstruct the image and has low linear probing accuracy, as the effective maximization of $I(z_i | X_i \cdot (1-m))$ requires a small reconstruction loss as per Assumption 3.
(3) Further adding $l_i^{(\text{min\_mi})}$ in (d) reduces the accuracy of (b) by 0.1 percentage points, since the approximation network needs to be trained by $\mathcal{L}_{\text{approx}}$ to predict the correct conditional relation of $p(z_i | X_i \cdot (1-m))$.
(4) Optimizing the complete losses of mutual information minimization boosts the accuracy by 0.2 percentage points in (e) and by 0.3 percentage points in our complete method (f).

**Loss weights.** We conduct experiments to tune the loss weights $\lambda_2$ and $\lambda_3$ for $l_{i,k}^{(\text{max\_mi})}$ and $l_i^{(\text{min\_mi})}$, respectively. For the loss weight $\lambda_1$ of the original reconstruction loss, we keep it at 1. As summarized in Tab. 3 (b), the optimal result occurs when $\lambda_2 = 1$ and $\lambda_3 = 10$.

**Orthogonal masks *vs.* independent masks.** In our method, we generate four orthogonal masks for an image, with each randomly covering 75% of the pixels, and the remaining 25% of the visible pixels from each mask do not overlap (*i.e.*, when combined, these masked images reveal the entire original image). We compare the results of the orthogonal masks and the independently generated masks. As summarized in Tab. 3 (c), the pre-training with independently generated masks drops the accuracy by 0.2%, suggesting that complete mutual information learning of all the image patches helps MI-MAE to learn better representations. This phenomenon makes sense since in the independent case, there is no guarantee that $X_0$ is 0. Thus, the second condition in Corollary 4 worsens compared to the orthogonal case.

**Masking ratios.** We design experiments to explore the influences of masking ratios. Unlike the original MAE, which generates one mask for each image, our MI-MAE generates 4 masks for each image with a masking ratio of 0.75. For other masking ratios, we design two masking strategies: (1) Complete masking: the number of orthogonal masks is determined by $\min(1/(1-\text{ratio}), 2)$. For the number $N$ lower than 2, we set it to 2 to utilize our losses. This ensures every iteration processes all the patches of each image. (2) Fixed 4 masks: all ratios use the same 4 masks. The training epochs are adjusted to match the same number of mask samples.

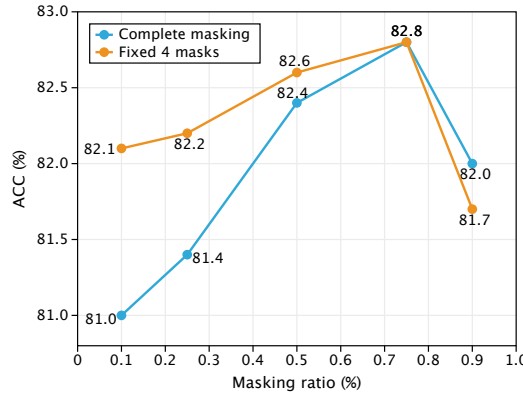

Figure 2: Ablation of masking ratios.

As summarized in Fig. 2, we observe optimal performance with a masking ratio of $0.75$. For other ratios, complete masking shows superiority at a ratio of $0.9$, while for smaller ratios, the fixed $4$ masks strategy gains advancements. The difference between these two strategies is that complete masking generates $10$ masks at a $0.9$ ratio, while only $2$ masks at ratios $0.1 \sim 0.5$. This indicates that more positive samples in MI maximization loss $l_{i,k}^{(\max_\text{m}\text{i})}$ leads to better performance. We also investigate the selected $0.75$ ratio with $6$ and $8$ masks, finding minor differences in the results ($82.9\%$, $82.7\%$). This is reasonable as we find the $l_j^{(\min\_\text{mi})}$ of all the numbers are close to $0$ and the third condition of Corollary 4 is well satisfied for each $N$. This means that the third condition of Corollary 4 only relies on $I(\hat{z}_k; X_j | z_i)$, which is already considered in $l_j^{(\min\_\text{mi})}$. For simplicity, we keep our complete masking strategy.

We also find that the reconstruction loss does not change much, while $l_j^{(\min\_\text{mi})}$ remains at a low level. This means $\epsilon_l$ in Assumption 3 can always be satisfied as observed in Appendix A.2.

**Pre-train with ViT-S.** To validate our efficacy on other model size, we perform pre-training on a smaller ViT-S model. As summarized in Tab. 4, with ViT-S, our MI-MAE obtains $79.8\%$ accuracy on ImageNet, which still obviously outperforms MAE by $1.4\%$. This further demonstrate the effectiveness of our method.

Table 4: ImageNet results on different ViTs.

| Model | Method | FT | LIN |
|---|---|---|---|
| ViT-B | MAE | 83.3 | 65.6 |
| | MI-MAE | 83.9 | 67.9 |
| ViT-S | MAE | 78.4 | 50.5 |
| | MI-MAE | 79.8 | 53.1 |

## 5.4 ARCHITECTURE OF APPROXIMATION NETWORK

Following CLUB (Cheng et al., 2020), we design an approximation network to estimate the posterior distribution of $p(\hat{z}_j | X_j)$. Similar to VAE works (Kingma & Welling, 2013; Pu et al., 2016), the approximation network has two branches to predict the mean $\mu(X_j)$ and variance $\sigma(X_j)$ of the Gaussian distribution, respectively. Then, the approximation is determined by $q_\theta(\hat{z}_j | X_j) = \mathcal{N}(\hat{z}_j; \mu(X_j), \sigma(X_j))$.

We use simple architectures in these two branches to predict the mean and variance. For the $\mu(X_j)$ branch, taking $X_j$ as input, we adopt a multi-layer perceptron (MLP) consisting of two fully-connected layers and an intermediate GELU activation function to encode the feature, then predict $\mu(X_j)$ by another fully-connected layer and a LeakyReLU activation. Similarly, the $\sigma(X_j)$ branch has a similar architecture but with the last activation LeakyReLU replaced by ReLU. We train the approximation network simultaneously with the autoencoder as in Algorithm 1.

**Note on the computation cost.** The approximation network is light-weight and is optimized by reusing the input $X$ and latent feature $\hat{z}$ obtained in autoencoder training. Hence, we did not observe noticeable increment in the training time.

## 6 CONCLUSION

In this paper, we introduced a new perspective for understanding and improving masked autoencoders (MAEs) by leveraging the information bottleneck (IB) theory. Building on these insights, we proposed MI-MAE, a novel method that enhances MAEs through mutual information maximization and minimization losses on the latent features. We conducted extensive theoretical and empirical analyses of our method, and experiments on tasks such as image classification, object detection, and semantic segmentation demonstrated its effectiveness. Our findings validate the theoretical framework and highlight the practical advantages of applying the information bottleneck principle to MAEs, providing deeper insights for developing more powerful self-supervised learning models. Future research could build on our findings to further explore and enhance the capabilities of MAEs, potentially leading to new advancements in self-supervised learning and computer vision.

## ACKNOWLEDGEMENTS

This work was supported in part by the Australian Research Council under Projects DP240101848 and FT230100549.

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

# A    APPENDIX

## A.1    PROOF OF THEOREM 2

### A.1.1    PROOF OF HOW EQUATION 4 INFLUENCE THE LAGRANGIAN TERM

From Eq. 3, to minimize the Lagrangian term, $I(X \cdot (1 - m); \widetilde{X \cdot (1 - m)})$ should be minimized. Since $I(\widetilde{X \cdot (1 - m)}; X \cdot (1 - m)) = I(X \cdot m; X \cdot (1 - m)) - (H(X \cdot m) - I(\widetilde{X \cdot (1 - m)}; X \cdot m))$. As $I(X \cdot m; X \cdot (1 - m))$ is determined only by the data, we minimize $I(\widetilde{X \cdot (1 - m)}; X \cdot m)$ which is the left-hand side of Eq. 4.

### A.1.2    PROOF OF BIAS IN THE ESTIMATION OF MUTUAL INFORMATION

According to the IB principle (Tishby & Zaslavsky, 2015), where $I(\hat{X}; Y) \leq \hat{I}(\hat{X}; Y) + O(\frac{K|y|}{\sqrt{n}})$, we derive the mutual information bound for the decoder of MAE. In this case, the decoder takes the latent feature $\widetilde{X \cdot (1 - m)}$ as the input and $X \cdot m$ as the output, so a generalization bound for the mutual information between the simplest effective description and the output as,

$$I(\widetilde{X \cdot (1 - m)}; X \cdot m) \leqslant \hat{I}(\widetilde{X \cdot (1 - m)}; X \cdot m) + O(\frac{K_x |Y|}{\sqrt{n_x}}). \tag{12}$$

However, it is hard to find the precise simplest effective description $\widetilde{X \cdot (1 - m)}$ due to the following reasons: (1) The computation of $\widetilde{X \cdot (1 - m)}$ is based on empirical data, which is influenced by sample size and distribution. This introduces biases and approximations into the optimization process. (2) The prediction of $\widetilde{X \cdot (1 - m)}$ is constrained by the model capacity of the encoder-decoder structure, which limits its ability to fully capture the optimal representation. As a result, in limited data distribution and model capacity, we can only find empirical estimation of biased simplest effective description as $\hat{I}(\hat{z}, ; X \cdot m)$, where $\hat{z} = \widetilde{X \cdot (1 - m)} + r$ and $r$ is the bias term.

We now provide formal proof to the existence of bias $r$. The mutual information between the optimal effective description $\widetilde{X \cdot (1 - m)}$ and the observed unmasked data $X \cdot m$ is defined as:

$$I(\widetilde{X \cdot (1 - m)}; X \cdot m) = H(\widetilde{X \cdot (1 - m)}) - H(\widetilde{X \cdot (1 - m)}|X \cdot m), \tag{13}$$

where $H$ is the entropy function. For the predicted latent feature $\hat{z}$, the empirical estimation of the mutual information is

$$\hat{I}(\hat{z}; X \cdot m) = \hat{H}(\hat{z}) - \hat{H}(\hat{z}|X \cdot m), \tag{14}$$

where $\hat{H}$ and $\hat{H}(\cdot|\cdot)$ are empirical estimates of entropy and conditional entropy, respectively.

We compare the true and empirical mutual information as

$$\Delta_{\mathrm{MI}} := \mathbb{E}[\hat{I}(\hat{z}; X \cdot m)] - I(X \cdot \widetilde{(1-m)}; X \cdot m)$$
$$= [\mathbb{E}[\hat{H}(\hat{z})] - H(X \cdot \widetilde{(1-m)})] - [\mathbb{E}[\hat{H}(\hat{z}|X \cdot m)] - H(X \cdot \widetilde{(1-m)}|X \cdot m)]. \tag{15}$$

As a result, the bias $r$ arises from two components:

1. Bias in the entropy term: $\mathbb{E}[\hat{H}(\hat{z})] - H(X \cdot \widetilde{(1-m)})$.

2. Bias in the conditional entropy term: $\mathbb{E}[\hat{H}(\hat{z}|X \cdot m)] - H(X \cdot \widetilde{(1-m)}|X \cdot m)$.

From the paper on entropy estimation (e.g., Paninski, 2003; Belghazi et al., 2018), we have

$$\mathbb{E}[\hat{H}(\hat{z})] - H(X \cdot \widetilde{(1-m)}) = O\left(\frac{K_z}{\sqrt{n_z}}\right), \tag{16}$$

where $K_z$ is the complexity of the latent space representation $\hat{z}$, and $n$ is the number of training samples. Similarly, for the conditional entropy term, we have

$$\mathbb{E}[\hat{H}(\hat{z}|X \cdot m)] - H(X \cdot \widetilde{(1-m)}|X \cdot m) = O\left(\frac{K_z}{\sqrt{n_z}}\right). \tag{17}$$

Combining the above results, the total bias in the estimation of mutual information can be bounded as $O\left(\frac{K_z}{\sqrt{n_z}}\right)$, which can prove that there indeed exists a bias.

### A.1.3 Proof of Generalization Bound via Biased Bottleneck

Unlike general discriminative networks, the target effective dimension of MAE is usually considered higher than the input effective dimension. Thus the bottleneck should be in the middle. The relevant information contained in the intermediate variables of the decoder is derived from the latent feature extracted by the encoder, despite the increment in the complexity. Intuitively, the latent feature is the information bottleneck in MAE. To further analyse the changes in effective information, we analyse the encoder and decoder of MAE separately.

Denote by $z$ the latent feature of the latent space, whose simplest effective description is denoted by $\hat{z}$. For the decoder, $z$ is the input and $(X \cdot m)$ is the ground truth. Denote the empirical estimation of mutual information by $\hat{I}(\cdot)$, and according to section 4.1, the upper bound on the input-ground truth mutual information is,

$$I(\hat{z}; X \cdot m) \leqslant \hat{I}(\hat{z}; X \cdot m) + O(\frac{K_z|X|}{\sqrt{n_z}}), \tag{18}$$

where $n_z$ is the size of the output, and $K_z$ is the complexity of the simplest effective description. In Eq. 18, the term $O(\frac{K_z|X|}{\sqrt{n_z}})$ describes the IB distortion, and goes worse with $K$. Since $I(X \cdot \widetilde{(1-m)}; X \cdot m) = I(\hat{z} - r; X \cdot m) = I(\hat{z}; X \cdot m) - I(\hat{z}; X \cdot m|r)$. We can expand the left-hand side of the formula 18 to get an upper bound of $I(X \cdot \widetilde{(1-m)}; X \cdot m)$ as

$$I(X \cdot \widetilde{(1-m)}; X \cdot m) \leqslant \hat{I}(\hat{z}; X \cdot m) + O(\frac{K_x|Y|}{\sqrt{n_x}}) - I(\hat{z}; X \cdot m|r), \tag{19}$$

It can be seen that this upper bound is better as the size of $z$ decreases. The decoder needs $z$ to capture as much information as possible, while the size of $z$ has to stay on the smallest possible scale. Thus, for decoder, the optimal case is that the simplest effective description of the hidden layer is the latent feature itself. Using the internal variables in encoder to take the place of latent feature, the decoder will suffer a worse generalization bound by a worse $O(\frac{K_z|X|}{\sqrt{n_z}})$.

### A.1.4 PROOF OF BIASED GENERALIZATION BOUND

This means that for a given masked image, there exists an unknown but optimal latent feature, denoted by $\zeta$. Assume that there exists an ideal mapping process from ground truth $X \cdot m$ to the optimal solution of the hidden layer as $\gamma(\cdot)$, the decoder, which takes latent feature as the input and approximates $X \cdot m$, can be interpreted as the inverse process of $\gamma(\cdot)$ (see the bottom part of Fig. 1), such that on a well optimized MAE, we can get an approximated latent $Z \approx \gamma(X \cdot m)$ from the decoder and $X \cdot m$, which satisfies

$$I(\zeta; X \cdot m) - I(Z; X \cdot m) = \epsilon_i, \tag{20}$$

where $\epsilon_i$ is a small constant.

Meanwhile, using the IB theory, taking the $X \cdot (1 - m)$ as the input of the encoder, whose simplest effective description is denoted by $\chi$, and $z$ as the predicted latent feature of the encoder, the generalization bound of the encoder is

$$I(\chi; \zeta) \leqslant \hat{I}(\chi; \zeta) + O(\frac{K_x |\zeta|}{\sqrt{n_x}}), \tag{21}$$

where $n_x$ denotes the size of the sample $X \cdot (1 - m)$, and $K_x$ denotes the length of the simplest effective description. Thus, considering both Eq. 20 and Eq. 21 and using the internal variables in decoder to take the place of latent feature, the encoder will also suffer a worse generalization bound by a worse $O(\frac{K_x |\zeta|}{\sqrt{n_x}})$. In conclusion, the latent feature is the simplest effective description for the whole MAE.

### A.2 PROOF OF ASSUMPTION 3

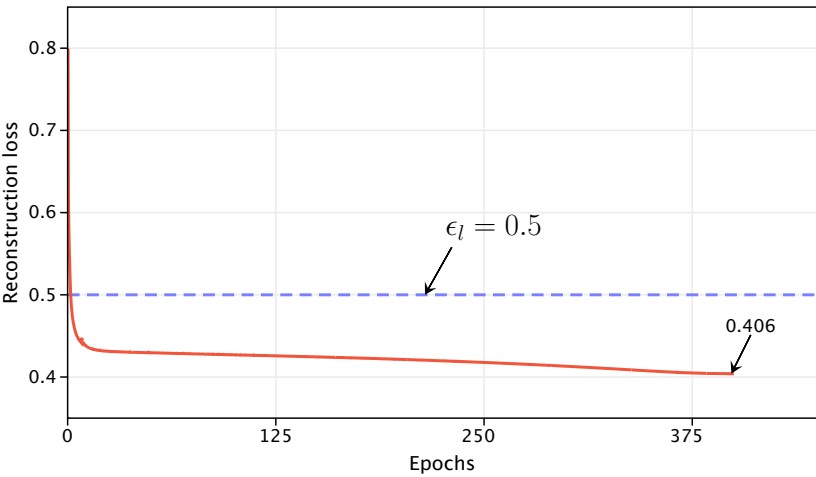

Figure 3: The curve of reconstruction loss $\mathcal{L}_{\text{rec}}$ during the 400-epoch training of MI-MAE. We set $\epsilon_l$ to 0.5.

Fig. 3 shows that $\mathcal{L}_{\text{rec}}$ satisfies Assumption 3. This means the $L_2$ distance between $Y$ and $\hat{Y}$ can be limited. The mutual information between $Y$ and $\hat{Y}$ is negatively relevant to the $L_2$ distance between them. Assume that when $\mathcal{L}_{\text{rec}} = \epsilon_l$, $I(Y; \hat{Y}) = \alpha$. When $\mathcal{L}_{\text{rec}} \leqslant \epsilon_l$, $I(Y; \hat{Y}) \geqslant \alpha$. When $Y$ is given, $H(Y|\hat{Y})$ can be calculated from $I(Y; \hat{Y})$ as,

$$H(Y|\hat{Y}) = H(Y) - I(Y; \hat{Y}). \tag{22}$$

where $H(Y)$ is fixed. Since $I(Y; \hat{Y}) \geqslant \alpha$, for $\epsilon_Y = H(Y) - \alpha$, we have $H(Y|\hat{Y}) \leqslant \epsilon_Y$

## A.3 PROOF OF COROLLARY 4

For a normal MAE consider $m_i \in M$, the mutual information between the prediction $\hat{Y}_i$ and the ground truth $Y_i$ can be denoted as $I(\hat{Y}_i; Y_i)$. We have,

$$
\begin{aligned}
&I(\hat{Y}_i; Y_i) \\
=&I(\hat{Y}_i; X_0 + \sum_{j=1; j \neq i}^{n} X_j) \\
\leqslant&I(\hat{Y}_i; X_0) + \sum_{j=1; j \neq i}^{n} I(\hat{Y}_i; X_j)
\end{aligned}
\tag{23}
$$

With Assumption 3, we can easily find another sample whose input is $X \cdot (1 - m_k)$ and output is $Y_k$, where $m_k \in M$ is orthogonal to $m_i$. THe mutual information between $\hat{Y}_k$ and $Y_k$ is,

$$
I(\hat{Y}_k; Y_k) \leqslant \iota_k + I(\hat{Y}_k; X_i),
\tag{24}
$$

where $\iota_k = I(\hat{Y}_k; X_0) + \sum_{n; j \notin \{i,k\}}^{j=1} I(\hat{Y}_k; X_j)$.

The upper bound described in Eq. 24 is approximate to the upper bound described in Eq. 23. The objectives of the other samples in $M$ differ from those of $i$ by less than $I(Y_k; X_i) - I(Y_i; X_k)$, which is also a small constant. Thus, for networks with the same initial state, the information about the intermediate variables in the network inference process is very close when sampling different masks in $M$. This shows the possibility of using latent feature of a neighbourhood as objective.

Encouraged by Theorem 2, we can write the similar mutual information on latent space. For and $k$ that satisfies $k \in \mathbb{Z} \cap [1, N]$ and $j \neq i$, the mutual information $I(\hat{z}_k; z_k)$ is,

$$
\begin{aligned}
&I(\hat{z}_k; z_k) \\
=&I(\hat{z}_k; \sum_{n; j \notin \{i,k\}}^{j=0} X_j) + I(X_i; \hat{z}_k) \\
\leqslant&I(\hat{z}_k; X_0) + \sum_{n; j \notin \{i,k\}}^{j=1} I(\hat{z}_k; X_j) + I(\hat{z}_k; \hat{z}_k) - \epsilon_z + I(\hat{z}_k, X_i).
\end{aligned}
\tag{25}
$$

Taking the known $z_i$ into Eq.25, the upper bound can then be rewritten as,

$$
\begin{aligned}
&I(\hat{z}_k; z_k) \\
\leqslant&[I(z_i, X_0) - I(\hat{z}_k, X_0|z_i)] + [\sum_{n; j \notin \{i,k\}}^{j=1} I(\hat{z}_i; X_j) - \sum_{n; j \notin \{i,k\}}^{j=1} I(\hat{z}_i; X_j|z_k)] + I(z_i; z_k). \\
=&I(\hat{z}_k; z_k) \leqslant l_i + I(\hat{z}_k; \hat{z}_i) - I(\hat{z}_i; X_0|z_k) - \sum_{n; j \notin \{i,k\}}^{j=1} [I(\hat{z}_i; X_j|z_k) - I(\hat{z}_k; X_j|z_i)].
\end{aligned}
\tag{26}
$$

Rearranging the above equation gives the equation in the corollary. $l_i$ is decided only by $I(\hat{z}_i; z_i)$. $I(\hat{z}_k, X_0|z_i)]$ is decided only by $X_0$ when $z_i$ is given. So a small $H(X_0)$ can help get bigger $I(\hat{z}_k; z_k)$. For any $j$ that satisfies $j \in \mathbb{Z} \cap [1, N]$ and $j \notin \{i, k\}$ $I(\hat{z}_i; X_j|z_k)]$, when $z_i$ is given, $I(\hat{z}_i; X_j)$ and $I(\hat{z}_i; X_j|z_i)$ is fixed, while $I(\hat{z}_k; X_j|z_k) \propto I(z_k; X_j) \propto I(z_k; z_j)$. Thus, the three conditions about maximizing $I(\hat{z}_k; z_k)$ given in Corollary 6 holds.

## A.4 DERIVATIONS OF THE UPPER BOUND OF MI

To minimize the mutual information between predicted latency $X_j$ and its corresponding input $X_j$, we leverage an upper bound of MI defined in CLUB (Cheng et al., 2020) is (we will show the proof

of this upper bound later):

$$I(X_j, \hat{z}_j) \leq \mathbb{E}_{p(X_j, \hat{z}_j)}[\log p(\hat{z}_j|X_j)] - \mathbb{E}_{p(X_j)p(\hat{z}_j)}[\log p(\hat{z}_j|X_j)]. \tag{27}$$

However, the conditional relation $p(\hat{z}_j|X_j)$ is intractable. We can instead use a variational distribution $q_\theta(\hat{z}_j|X_j)$ parameterized by $\theta$ to approximate it. Consequently, the upper bound $\hat{I}(X_j, \hat{z}_j)$ of mutual information in Eq. 27 becomes:

$$\hat{I}(X_j, \hat{z}_j) := \mathbb{E}_{p(X_j, \hat{z}_j)}[\log q_\theta(\hat{z}_j|X_j)] - \mathbb{E}_{p(X_j)p(\hat{z}_j)}[\log q_\theta(\hat{z}_j|X_j)]. \tag{28}$$

Nevertheless, $\hat{I}(X_j, \hat{z}_j)$ in Equation 28 no longer guarantees an upper bound of $I(X_j; \hat{z}_j)$ due to the variational approximation. Fortunately, we can prove that $\hat{I}(X_j, \hat{z}_j)$ can be a reliable upper bound estimator when the difference between $p(X_j, \hat{z}_j)$ and $q_\theta(X_j, \hat{z}_j)$ is small.

We first compare the difference between them as

$$\Delta := I(X_j, \hat{z}_j) - \hat{I}(X_j, \hat{z}_j), \tag{29}$$

With $H(X_j)$ being the entropy of variable $X_j$, and using the Mutual Information Neural Estimator (MINE) (Belghazi et al., 2018), we can rewrite mutual information $I(X_j, \hat{z}_j)$ as

$$\begin{aligned} I(X_j; \hat{z}_j) &:= H(X_j) - H(X_j|\hat{z}_j) \\ &= \mathbb{E}_{p(X_j, \hat{z}_j)}[\log p(\hat{z}_j|X_j) - \log p(\hat{z}_j)]. \end{aligned} \tag{30}$$

Therefore, with $q_\theta(X_j, \hat{z}_j) = q_\theta(\hat{z}_j|X_j)p(X_j)$ being the variational joint distribution induced by $q_\theta(\hat{z}_j|X_j)$, Eq. 29 can be reformulated by Eq. 30 and Eq. 28 as

$$\begin{aligned} \Delta :=&I(X_j, \hat{z}_j) - \hat{I}(X_j, \hat{z}_j) \\ =&\mathbb{E}_{p(X_j, \hat{z}_j)}[\log p(\hat{z}_j|X_j) - \log p(\hat{z}_j)] - \mathbb{E}_{p(X_j, \hat{z}_j)}[\log q_\theta(\hat{z}_j|X_j)] + \mathbb{E}_{p(X_j)p(\hat{z}_j)}[\log q_\theta(\hat{z}_j|X_j)] \\ =&\mathbb{E}_{p(X_j, \hat{z}_j)}[\log p(\hat{z}_j|X_j) - \log q_\theta(\hat{z}_j|X_j)] - \mathbb{E}_{p(X_j, \hat{z}_j)}[\log p(\hat{z}_j)] + \mathbb{E}_{p(X_j)p(\hat{z}_j)}[\log q_\theta(\hat{z}_j|X_j)] \\ =&\mathrm{KL}(p(X_j, \hat{z}_j)||q_\theta(X_j, \hat{z}_j)) - \mathrm{KL}(p(X_j)p(\hat{z}_j)||q_\theta(X_j, \hat{z}_j)). \end{aligned} \tag{31}$$

The above equation shows that, **(1)** when $\mathrm{KL}(p(X_j, \hat{z}_j)||q_\theta(X_j, \hat{z}_j)) \leq \mathrm{KL}(p(X_j)p(\hat{z}_j)||q_\theta(X_j, \hat{z}_j))$, we can directly get $I(X_j, \hat{z}_j) \leq \hat{I}(X_j, \hat{z}_j)$, and $\hat{I}(X_j, \hat{z}_j)$ is already an upper bound of MI. **(2)** Otherwise, if $\mathrm{KL}(p(X_j, \hat{z}_j)||q_\theta(X_j, \hat{z}_j)) > \mathrm{KL}(p(X_j)p(\hat{z}_j)||q_\theta(X_j, \hat{z}_j))$, by learning a good variational approximation $q_\theta(X_j, \hat{z}_j)$ that closes to $p(X_j, \hat{z}_j)$, we have minimized $\mathrm{KL}(p(X_j, \hat{z}_j)||q_\theta(X_j, \hat{z}_j)) < \epsilon_q$, then $|\hat{I}(X_j, \hat{z}_j) - I(X_j, \hat{z}_j)| < \epsilon_q$, $\hat{I}(X_j, \hat{z}_j)$ can become an MI estimator whose absolute error is bounded by the approximation performance $\mathrm{KL}(p(X_j, \hat{z}_j)||q_\theta(X_j, \hat{z}_j))$.

**Derivation of $\mathcal{L}_{\mathrm{approx}}$ in Eq. 9.** We show that $\mathrm{KL}(p(X_j, \hat{z}_j)||q_\theta(X_j, \hat{z}_j))$ can be minimized by minimizing the negative log-likelihood of $q_\theta(\hat{z}_j, X_j)$, because of the following equation:

$$\begin{aligned} &\min_\theta \mathrm{KL}(p(X_j, \hat{z}_j)||q_\theta(X_j, \hat{z}_j)) \\ =&\min_\theta \mathbb{E}_{p(X_j, \hat{z}_j)}[\log(p(\hat{z}_j|X_j)p(X_j)) - \log(q_\theta(\hat{z}_j|X_j)p(X_j))] \\ =&\min_\theta \mathbb{E}_{p(X_j, \hat{z}_j)}[\log p(\hat{z}_j|X_j)] - \mathbb{E}_{p(X_j, \hat{z}_j)}[\log q_\theta(\hat{z}_j|X_j)]. \end{aligned} \tag{32}$$

Eq. 32 equals to maximizing the second term $\max_\theta \mathbb{E}_{p(X_j, \hat{z}_j)}[\log q_\theta(\hat{z}_j|X_j)]$, as the first term has no relation to $\theta$, and hence the learning object $\mathcal{L}_{\mathrm{approx}}$ of $\theta$ is

$$\mathcal{L}_{\mathrm{approx}} = \frac{1}{N} \sum_{j=1}^{N} -\log q_\theta(\hat{z}_j|X_j). \tag{33}$$

**Derivation of $l_j^{(\mathrm{min\_mi})}$ in Eq. 10.** The MI upper bound $\hat{I}(X_j; \hat{z}_j)$ has an unbiased estimation as

$$\hat{I}(X_j, \hat{z}_j) = \log q_\theta(\hat{z}_j|X_j) - \frac{1}{N} \sum_{k=1}^{N} \log q_\theta(\hat{z}_k|X_j), \tag{34}$$

which reflects our $l_j^{(\mathrm{min\_mi})}$ in Eq. 10.

## A.5 ROBUSTNESS EVALUATION

Compared to MAE, our method with explicit mutual information optimization based on information bottleneck, would have better robustness, as the IB helps guide latent features to suppress noise while retaining semantic information. To validate this, we test our fine-tuned model on two popular ImageNet variants for robustness evaluation, ImageNet-A (Hendrycks et al., 2021) and ImageNet-C (Hendrycks & Dietterich, 2019). As summarized in Tab. 5, we report the top-1 accuracy on ImageNet-1K and ImageNet-A, and mean corruption error (mCE, lower is better) on ImageNet-C. The results show that, our MI-MAE significantly improves the performance of MAE, indicating better robustness.

Table 5: Robustness evaluation results.

| Method | ImageNet-1K ACC | ImageNet-A ACC | ImageNet-C mCE |
|--------|-----------------|----------------|----------------|
| MAE | 83.3 | 35.9 | 51.7 |
| MI-MAE | 83.9 | 37.4 | 49.5 |

