# OpenReview forum: "Learning Mask Invariant Mutual Information for Masked Image Modeling"
_ICLR.cc/2025/Conference — ICLR 2025 Poster_

### Official Review · Reviewer_3dfm · 2024-10-27

**Soundness:** 3
**Presentation:** 3
**Contribution:** 3
**Rating:** 6
**Confidence:** 5

**Summary:**

This paper proposes to interpret Masked Autoencoders (MAE) using information bottleneck principle. It first conduct a theoretical analyses to show that balance the relevant and irrelevant information in the latent features is the key to improve MAE performance. Then it introduces an improved MI-MAE by maximizing the relevant information between input and latent features and minimize the relevant information between output and latent features. This is achieved by introducing two new losses functions beside the original reconstruction loss. Experiments on image classification, object detection and segmentation reveal the effectiveness of the proposed MI-MAE

**Strengths:**

+ This paper provides a new perspective in understanding the MAE using information bottleneck in information theory, which distinguishes itself with other methods. Moreover, this paper provide detailed proof on how to understand MAE and how to improve MAE with the idea of information bottleneck.

+ The paper introduces two types of mutual information based losses on the latent space. This is derived and supported by the theoretical  proof.

+ Validation on various experiments show the effectiveness of MI-MAE. Specifically, MI-MAE can achieve better results compared with MAE even with 4X fewer pretraining epochs. The method can also be generalized to other mask image modeling method such as SimMIM.

**Weaknesses:**

- The paper mentions that during each training iteration, it has 4 masks for each image, which can be regraded as data augmentation during training. For a fair comparison with MAE 400 epoch, it might be worth trying to run MAE with 4 augmented masks but without the proposed information losses to truly ablate the effectiveness of the information losses. To be more specific, we could compare (1) Standard MAE (already have). (2). MAE with 4 masks per image but no information losses (3). Full MI-MAE with 4 masks and information losses(already have).

**Questions:**

- As demonstrated in Table 1, the MI-MAE generally performs better in linear probing (LIN) or fine-tuning with 1% (FT1%) of the data compared to full fine-tuning (FT). Could the authors provide more details on why this occurs from the perspective of information bottleneck?

- In the caption of Figure 1 Line 177-178, why the notions of two losses l(max_mi) and l(min_mi) doesn't follow \mathcal{L}_{\text{rec}} as the same format? I would recommend  standardizing the notation for consistency, if there isn't a specific reason for the difference.

---

> ### Author Response · Authors · 2024-11-21
> **Authors' Response to Reviewer 3dfm**
>
> Dear Reviewer 3dfm,
>
> We sincerely thank you for your valuable and constructive comments. Below, we address your comments and questions in detail.
>
> > **Q1: Comparison to MAE with 4 masks per image but no information losses.**
>
> **R1:** Thank you for the suggestion! We have conducted this comparison in Table 3 of the ablation study. Specifically, the combination (a), which uses the same input data, batch size, and number of iterations as our MI-MAE but only applies the original MAE loss, achieves 0.6% lower accuracy compared to our full MI-MAE (f). This demonstrates the added value of our information-based objectives beyond the influence of input samples or training strategy.
>
> ---
>
> > **Q2: MI-MAE generally performs better in linear probing (LIN) or fine-tuning with 1% (FT1%) of the data compared to full fine-tuning (FT). Could the authors provide more details on why this occurs from the perspective of information bottleneck?**
>
> **R2:** Thank you for your question. We believe the more noticeable improvements on LIN and FT-1% are not solely due to the information bottleneck principle, as similar trends are observed in other methods like MFF. Our explanations are as follows:
> 1. LIN and FT-1% generally have much lower accuracies than FT, making it easier to achieve more significant relative improvements.
> 2. LIN and FT-1% rely more heavily on the quality of pre-training due to their limited training data. Methods with explicit feature regularizations, like MI-MAE, tend to have a greater impact under these settings, as they produce more robust pre-trained representations.
> 3. The IB principle explicitly encourages the retention of relevant information while discarding irrelevant information. This leads to pre-trained features that are more compact and robust.
>
> ---
>
> > **Q3: Standardize the notation for consistency.**
>
> **R3:** Thank you for your helpful suggestion. We differentiate $\mathcal{L}_\text{rec}$ and $l$, as $\mathcal{L}$ represents the overall loss for a batch, while a $l$ denote the loss for an image pair in MI maximization or an image in MI minimization. We have revised the paper to unify the notations by adding the batch-level losses.

---

> > ### Comment · Reviewer_3dfm · 2024-11-27
> > **Thanks for your responce.**
> >
> > Thanks for authors' response. It has resolved all my qeustions and I've raised by confidence from 3 to 5.

---

### Official Review · Reviewer_PnEF · 2024-11-01

**Soundness:** 3
**Presentation:** 3
**Contribution:** 3
**Rating:** 6
**Confidence:** 3

**Summary:**

This paper proposed MI-MAE, extending the MAE framework by maximizing relevant and minimizing irrelevant information in latent representations. MI-MAE introduces mutual information-based losses in the encoder's latent space to enhance feature representation. The method optimizes two main loss types, maximizing mutual information across orthogonal masks to retain relevant information and minimizing mutual information between the input and latent space to filter out unnecessary data. Additionally, MI-MAE's setup includes generating multiple orthogonal masks per image, which are reconstructed to validate the relevant mutual information content across different patches. Experiments reveal that MI-MAE outperforms standard MAE configurations across some benchmarks, including ImageNet and COCO.

**Strengths:**

1. This work provides some new perspective in analyzing MAE with MI backed motivations, and resulting improvements demonstrated the practicability of applying mutual information maximization and minimization within latent representations and between inputs.
2. The paper demonstrates MI-MAE’s efficacy across a variety of vision tasks, including image classification, object detection, and semantic segmentation. In reported results, MI-MAE shows better efficiency in terms of number of training epoch comparing to MAE with comparable accuracy.
3. This paper provides detailed ablations on effects of different components, such as mask generation strategies, loss functions, and loss weight configurations.

**Weaknesses:**

1. Increased complexity in training. Additional loss terms $l^{max_mi}$ and $l^{min_mi}$ requires weighting parameters introduced (i.e. $\lambda_1,\lambda_2, \lambda_3$) which are empirically determined, this makes the optimization more complicated than vanilla MAE.
2. This method uses an approximation network to estimate variational distributions for mutual information minimization. It also introduces another layer of approximation, which may not capture the true complexity of the mutual information in the latent space accurately. This could lead to sub-optimal representation learning if the approximation fails.
3. This paper assumes that the model can effectively minimize information distortion in intermediate layers as data progresses through the encoder-decoder structure, which might lead to the over-compression relevant information.
4. Lack of robustness analysis. An important aspect of using information bottleneck is that it can increase the robustness of the pre-trained model. It is of interest to test out how this method performs in ImageNet-A/C validation set.

**Questions:**

Please refer to the Weakness section.

---

> ### Author Response · Authors · 2024-11-21
> **Authors' Response to Reviewer PnEF**
>
> Dear Reviewer PnEF,
>
> We sincerely thank you for your effort in reviewing our paper. Your insightful and helpful comments helped us for refining our paper better. Below, we address your comments and questions in detail.
>
> > **Q1: Loss weights are empirically determined.**
>
> **R1:** The MI maximization and minimization losses have different implementations and objectives. To achieve an optimal balance between these terms, we conducted experiments to empirically determine suitable values for the weights $\lambda_2$ and $\lambda_3$. However, we set $\lambda_1$ to 1 for all experiments, and as shown in Table 3 (b), changing $\lambda_2$ and $\lambda_3$ does not lead to significant performance differences. Notably, even the default configuration (all weights set to 1) performs better than baseline MAE. Therefore, one can either use the default weights (all 1s) or select weights based on our experimental results to achieve strong performance. This process is straightforward and does not introduce significant complexity.
>
> ---
>
> > **Q2: The approximation network for mutual information minimization may not capture the true complexity of the mutual information in the latent space accurately.**
>
> **R2:** Thank you for your comment. Similar to prior works like CLUB (Cheng et al., 2020) and VAE (Kingma & Welling, 2013; Pu et al., 2016), we use a simple MLP-based approximation network. These methods have shown that such networks can perform well for mutual information estimation. In our work, we also find empirically that the simple network is sufficient to estimate the mean and variance of the input accurately, enabling effective mutual information minimization and strong overall performance. While more complex networks could be explored, our current design achieves a good balance between simplicity and effectiveness.
>
> ---
>
> > **Q3: This paper assumes that the model can effectively minimize information distortion in intermediate layers as data progresses through the encoder-decoder structure, which might lead to the over-compression relevant information.**
>
> **R3:** Thank you for raising this question. As shown in Theorem 2, our method does not reduce the total amount of information. Specifically, in Equation (4), our approach minimizes $I(\hat{z}; X \cdot m | r)$, which limits the irrelevant information, without affecting $\hat{I}(\hat{z}; X \cdot m)$, the total mutual information. Therefore, minimizing information distortion does not lead to the compression of relevant information.
>
> ---
>
> > **Q4: Lack of robustness analysis. An important aspect of using information bottleneck is that it can increase the robustness of the pre-trained model. It is of interest to test out how this method performs in ImageNet-A/C validation set.**
>
> **R4:** Thank you for your valuable suggestion. We have included a robustness analysis in Section A.5 of the Appendix, and here is a summary of the results:
> |Method|ImageNet-1K ACC |ImageNet-A ACC |ImageNet-C mCE (lower is better)|
> |:--:|:--:|:--:|:--:|
> |MAE|83.3|35.9|51.7|
> |MI-MAE|**83.9**|**37.4**|**49.5**|
>
> We evaluated fine-tuned models on ImageNet-A and ImageNet-C. The results show that MI-MAE significantly improves the robustness of MAE, with higher accuracy on ImageNet-A and lower mCE on ImageNet-C. This demonstrates that explicitly applying the information bottleneck principle helps guide latent features to suppress noise while retaining semantic information.

---

> > ### Comment · Reviewer_PnEF · 2024-11-27
> >
> > I appreciate author's detailed response and I would maintain my current score.

---

### Official Review · Reviewer_MLQd · 2024-11-04

**Soundness:** 3
**Presentation:** 3
**Contribution:** 3
**Rating:** 8
**Confidence:** 4

**Summary:**

This paper studied Masked Autoencoders via information bottleneck: MAE can be interpreted as obtaining the simplest effective distortion to capture all information between masked and recovered images. Then it proposes to add two losses to improve MAE: maximize the mutual information between the latents of different masked views of the same image, and also minimize the mutual information between the latents and the input. The authors show a 0.5% improvement on ImageNet-1K with 400-epoch training compared to 1600-epoch MAE, and outperforms MAE on transfer tasks such as instance segmentation and semantic segmentation.

**Strengths:**

Originality: this is the first paper to study MAE under information bottleneck. It re-interprets MAE as minimizing a Lagrangian term that includes two terms: the simplest effective description of input and distortion of the network. It then argues that the MAE can only find a suboptimal solution. Finally it introduced two terms, based on explicit terms (MI maximization and minimization) to enforce IB principle, to improve MAE.

Significance: the knowledge of interpreting MAE from an IB viewpoint is useful for the community.

**Weaknesses:**

Clarity:

The overall motivation is clear, but many small explanations are missing.

(a) First, it is unclear from the discussions after Eq. (3), why MAE can only find a sub-optimal effective description. The reviewer would appreciate more explanations, such as specific constraints or limitations, preferably with formal proofs, that prevent MAE from finding the optimal solution

(b) Sorry if the reviewer has missed it, but it is not clear, after Eq. (4), why mitigating the bias $r$ would help MAE. What is the exact effect of $r$ on the upper bound? Why is improving that upper bound helpful for the LHS of Eq. (4)? And how does the LHS of Eq. (4) directly impact MAE, as the LHS in Eq. (4) is neither the $D_{IB}$ term nor the first MI term in RHS of Eq. (3)? Any mathematical derivation showing these would be very helpful.

(c) Following up on the last question, there is no clear theoretical link to how the proposed losses can *directly* improve Eq. (3). It is briefly explained in Lines 226 - 227, that “maximizing the mutual information between the latent feature and $\zeta$ will help reduce $I(\hat{z}; X \cdot m | r)$”, but why? Are there any possible direct proofs? Even though this is true, how does reducing $I(\hat{z}; X \cdot m | r)$ directly affect the IB formulation in Eq. (3)? Any concrete steps showing these would be very helpful.

Quality:

The empirical improvement is unfortunately not substantial.

**Questions:**

Where is the proof of Eq. (12)? There is no clear description of the IB distortion term in Eq. (12); how did the authors derive such an error bound, by what theorems? There is the same issue for Eq. (14). Detailed proofs by stating the theorems used will be helpful.

MINE is known for high variance; have the authors considered other alternative estimators?

---

> ### Author Response · Authors · 2024-11-21
> **Authors' Response to Reviewer MLQd (Part 1)**
>
> Dear Reviewer MLQd,
>
> The authors sincerely than you for your insightful comments and constructive feedback. Below, we address your comments and questions in detail.
>
> > **Q1: It is unclear from the discussions after Eq. (3), why MAE can only find a sub-optimal effective description.**
>
> **R1:** MAE can only find a sub-optimal (biased) effective description $\widetilde{X \cdot (1-m)} + r$ due to the following reasons:
> 1. The computation of $\widetilde{X \cdot (1-m)}$ is based on empirical data, which is influenced by sample size and distribution. This introduces biases and approximations into the optimization process.
> 2. The prediction of $\widetilde{X \cdot (1-m)}$ is constrained by the model capacity of the encoder-decoder structure, which limits its ability to fully capture the optimal representation.
>
> As a result, in limited data distribution and model capacity, we can only find empirical estimation of biased simplest effective description as $\hat{I}(\hat{z}, ; X \cdot m)$, where $\hat{z} = \widetilde{X \cdot (1-m)} + r$ and $r$ is the bias term.
>
> We have added this detailed explanation into Section A.1.1 of our revision.
>
> ---
>
> > **Q2: In Eq. (4), why mitigating the bias r would help MAE. What is the exact effect of r on the upper bound? Why is improving that upper bound helpful for the LHS of Eq. (4)? And how does the LHS of Eq. (4) directly impact MAE, as the LHS in Eq. (4) is neither the DIB term nor the first MI term in RHS of Eq. (3)?**
>
> **R2:**
>
> 1. **Why mitigating $r$ helps MAE:**
> In Theorem 2, the bias $r$ is introduced in the sub-optimal effective description $\widetilde{X \cdot (1-m)} + r$, this bias reflects the deviation between the learned latent representation and the true optimal effective description. Reducing $r$ improves the alignment between the latent feature $\hat{z}$ and the true effective description $\widetilde{X \cdot (1-m)}$, leading to more accurate representations and reduced information distortion.
> 2. **Effect of $r$ on the upper bound:**
> In Eq. (4), $I(\widetilde{X \cdot (1-m)}; X \cdot m)$ is upper-bounded by the empirical estimate $\hat{I}(\hat{z}; X \cdot m)$, an error term $O(\frac{K_x|Y|}{\sqrt{n_x}})$, and a penalty $- I(\hat{z};X \cdot m|r)$. The bias $r$ affects this penalty term, as a larger $r$ reduces the conditional mutual information $I(\hat{z};X \cdot m|r)$. In formula, it is $r \propto -I(\hat{z};X \cdot m|r) $.  By mitigating $r$, this penalty $- I(\hat{z};X \cdot m|r)$ is minimized, tightening the upper bound and improving the overall information estimation.
> 3. **Why improving the upper bound helps the LHS of Eq. (4):**
> The LHS of Eq. (4), $I(\widetilde{X \cdot (1-m)}; X \cdot m)$, represents the mutual information between the latent representation and the unmasked parts of the image. By tightening the upper bound, the LHS of Eq. (4) can be more accurately estimated via the empirical mutual information $\hat{I}(\hat{z}; X \cdot m)$ and the bias penalty $- I(\hat{z};X \cdot m|r)$.
> 4. **Impact of the LHS of Eq. (4) on MAE:**
> While the LHS of Eq. (4) is not explicitly the $D_{IB}$ term or the first MI term in Eq. (3), it implicitly affects both. In formula, we have
> $I(\widetilde{X \cdot (1-m)}, X \cdot (1-m)) = I(X \cdot m, X \cdot (1-m))  \cap I(\widetilde{X \cdot (1-m)}, X \cdot m)$.
> As $I(X \cdot m, X \cdot (1-m))$ is determined by the dataset, $I(\widetilde{X \cdot (1-m)}, X \cdot m)$, which is the LHS of Eq. (4), can influence the Lagrange term in Eq. (3).

---

> ### Author Response · Authors · 2024-11-21
> **Authors' Response to Reviewer MLQd (Part 2)**
>
> > **Q3: there is no clear theoretical link to how the proposed losses can directly improve Eq. (3). It is briefly explained in Lines 226 - 227, that “maximizing the mutual information between the latent feature and ζ will help reduce $I(\hat{z}; X \cdot m | r)$”, but why? Are there any possible direct proofs? Even though this is true, how does reducing $I(\hat{z}; X \cdot m | r)$ directly affect the IB formulation in Eq. (3)?**
>
> **R3:** Thank you for this important question. Below, we provide detailed explanations and concrete steps to address your concerns:
>
> 1. **Why does maximizing mutual information reduce information distortion?**
> The information distortion term is defined as the gap between the ground truth (GT) latent feature $\zeta$ and the predicted latent feature $\hat{z}$. Since the GT latent feature $\zeta$ is fixed, increasing the mutual information between $\hat{z}$ and $\zeta$ ensures that more relevant information from $\zeta$ is retained in $\hat{z}$, thereby reducing the gap (distortion).
> **Proof:**
> $I(\hat{z} ;X⋅m|r) = I(\zeta; X⋅m) - I(\hat{z}; X \cdot m).$
> In lines 226 - 227, we consider the situation at a fixed $r$. In this case, as $\zeta$ is the optimal latent feature, we give that $I(\zeta; X⋅m)$ is fixed. To minimize $I(\zeta; X⋅m|r)$, we concentrate on maximizing the term $I(\hat{z}; X \cdot m)$.
> We have $I(\hat{z}; X \cdot m) = I(\zeta; X⋅m) \cap I(\zeta, \hat{z})$.
> As $I(\zeta; X⋅m)$ is fixed, increasing $I(\zeta, \hat{z})$ helps to increase $I(\hat{z}; X \cdot m)$, which decreases $I(\hat{z} ;X⋅m|r)$.
>
> 2. **How does reducing $I(\hat{z}; X \cdot m|r)$ affect Eq. (3)?**
> Minimizing the distortion will directly minimize the second subterm in Eq. (3), which leads to the decrease of Eq. (3).
>
> ---
>
> > **Q4: The empirical improvement is unfortunately not substantial.**
>
> **R4:** MI-MAE achieves consistent improvements over all the benchmarks. While the improvement on ImageNet fine-tuning may seem modest at 0.6%, it is double the gain reported by MFF (CVPR 2023), which achieved only 0.3%. Moreover, in tasks such as linear probing and 1% fine-tuning where pre-training quality is more critical, MI-MAE demonstrates substantial improvements of over 2% with 800-epoch pre-training. These results highlight the effectiveness and robustness of our method compared to previous approaches.
>
> ---
>
> > **Q5: Where is the proof of Eq. (12)? There is no clear description of the IB distortion term in Eq. (12); how did the authors derive such an error bound, by what theorems? There is the same issue for Eq. (14).**
>
> **R5:** Thank you for raising this point. We have explained Eq. (12) and Eq. (14) more detailedly in Section A.1 of our revision. Below, we provide clarifications regarding the proofs of Eq. (12) and Eq. (14):
> 1. **Proof of Eq. (12):** Based on the IB framework (Shamir et al., 2010), where $I(\hat{X}; Y) \le \hat{I}(\hat{X}; Y) + O(\frac{K|y|}{\sqrt{n}})$, we derive the mutual information bounds for the decoder of MAE. In this case, the decoder takes the latent feature $\hat{z}$ as input and outputs the masked image $X \cdot m$. Applying the IB principle to the decoder, we can rewrite the equation as Eq. (12).
> 2. **Proof of Eq. (14):** For the encoder, the optimal output that perfectly matches the decoder is considered the ground truth latent feature $\zeta$. Following IB (Shamir et al., 2010), taking $X \cdot (1-m)$ as the input and $\zeta$ as the output of the encoder, the mutual information bound can be expressed as Eq. (14).
>
> ---
>
> > **Q6: MINE is known for high variance; have the authors considered other alternative estimators?**
>
> **R6:** Thank you for highlighting this important aspect. We acknowledge that MINE (Mutual Information Neural Estimator) is known for its high variance. While we are open to exploring alternative estimators in future work, we chose MINE for the following reasons:
> 1. MINE is a well-established estimator within the context of IB theory. Its theoretical properties align well with the derivations and proofs used in our work, particularly in the context of MAE optimization.
> 2. Our primary contribution lies in the theoretical understanding of MAE through the lens of the information bottleneck principle and the introduction of novel objectives. The choice of a specific estimator, while important, is not the central focus of our work. Thus, we prioritized an estimator with a strong theoretical foundation, such as MINE, to support the proofs of our theorems.
>
> We recognize the potential benefits of exploring alternative estimators, such as variational estimators, which may mitigate the variance issues associated with MINE. We appreciate your suggestion and will consider more advanced and robust estimators in our future work.

---

> ### Comment · Reviewer_MLQd · 2024-11-23
> **Thank you for the response**
>
> The reviewer truly appreciates the authors' detailed response. Some of them helped clarify the paper. However, the reviewer cannot recommend an acceptance given my current understanding of the updated draft. The reviewer meant to ask for concrete derivations for these claims from the paper:
>
> R1: Any direct proof showing that the estimation is indeed biased (e.g., via expected value)?
>
> R2, 1: How does $I(\hat{z}; X \cdot m | r)$ come about in Thereom 2? The existing proof in A.1.1 did not show it. **This is an important missing step.**
>
> R2, 4: Thanks for discussing these implicit influences; however, is there any formal quantification of such influences? If the authors make such a claim, "mitigating the bias on the information bottleneck helps in achieving better MAE performance". formal proofs should be provided. Also, mutual information terms are scalars, intersections of two scalars may not make sense. Maybe the authors could consider conditional MI terms to make it more rigorous.
>
> R3: Why $I(\hat{z}; X \cdot m | r) = I(\zeta; X \cdot m) - I(\hat{z}; X \cdot m)$? Why consider a fixed $r$, when $r$ is unknown and model dependent, and do the authors intend to add this assumption in the paper? The same issue with the intersection of two mutual information, conditional MI terms may help clarify. Why $I(\hat{z}; X \cdot m) = I(\zeta; X\cdot m) \cap I(\zeta, \hat{z})$?
>
> R5: Could the authors point out any results from Shamir et al. 2010 that directly show $I(\hat{X}; Y) - \hat{I}(\hat{X}; Y) \leq O(\frac{K |y|}{\sqrt{n}})$? The only results related only shows $ | I(\hat{X}; Y) - \hat{I}(\hat{X}; Y) | < O(|y|)$, instead of  $I(\hat{X}; Y) - \hat{I}(\hat{X}; Y)  < O(|y|)$. **This is an important missing step.**
>
> The reviewer again appreciates the authors' tremendous effort in the response.

---

> > ### Author Response · Authors · 2024-11-26
> > **Follow-Up Response (Part 1)**
> >
> > Dear Reviewer MLQd,
> >
> > Thank you for your thoughtful feedback and for appreciating our detailed responses. We deeply value your constructive comments, which have been instrumental in further refining our work. Below, we address your latest concerns point by point:
> >
> > > **R1: Any direct proof showing that the estimation is indeed biased (e.g., via expective value)?**
> >
> > We appreciate your request for a direct proof. As detailed in the revised Appendix A.1.2, we provide an additional derivation that explicitly demonstrates the existence of bias in terms of the expected value. Specifically, we quantify the expected deviation between the estimated $\hat{I}(\hat{z}; X \cdot m)$ and the true $I(\widetilde{X\cdot (1-m)}; X \cdot m)$. The bias is formally shown to be bounded by the sample size and model complexity, with the error term scaling as $O(K_z / \sqrt{n_z})$. This derivation rigorously supports the claim that the estimation is inherently biased.
> >
> > ---
> >
> > > **R2, 1: How does $I(\hat{z};X \cdot m|r)$ come about in Thereom 2?**
> >
> > Thank you for pointing out this missing step. We have revised our proof by adding the following steps:
> > The mutual information $I(\widetilde{X \cdot (1-m)}; X \cdot m)$ can be expressed as:
> >
> > $I(\widetilde{X \cdot (1-m)}; X \cdot m)=I(\hat{z}-r;X \cdot m) = I(\hat{z};X \cdot m) - I(\hat{z};X \cdot m|r)$.
> >
> > Using this relation, we expand the left-hand side of the equation (18) to get an upper bound for $I(\widetilde{X \cdot (1-m)}; X \cdot m)$:
> >
> > $I(\widetilde{X \cdot (1-m)}; X \cdot m) \leqslant \hat{I}(\hat{z}; X \cdot m) + O(\frac{K_x|Y|}{\sqrt{n_x}}) - I(\hat{z};X \cdot m|r)$.
> >
> > ---
> >
> > **R2, 4:**
> >
> > > **1. Intersections of mutual information.**
> >
> > Thanks for pointing out this incorrect usage of intersections. We revise the equation in our response as $I(\widetilde{X \cdot (1-m)};X\cdot(1-m))=I(X\cdot m;X\cdot(1-m))-(H(X\cdot m)-I(\widetilde{X\cdot(1-m)};X\cdot m))$.
> >
> > > **2. Is there any formal quantification of the implicit influences discussed?**
> >
> > From this revised equation, increasing $I(\widetilde{X\cdot(1-m)};X\cdot m))$ directly increases $I(\widetilde{X \cdot (1-m)};X\cdot(1-m))$, which is the mutual information between the optimal solution and the ground truth.
> >
> > > **3. Proof of claim “mitigating the bias on the information bottleneck helps in achieving better MAE performance”.**
> >
> > With a limited training data distribution, finding the optimal solution $I(\widetilde{X\cdot(1-m)};X\cdot m))$ is challenging. Instead, we compute the empirical estimation $\hat{I}(\hat{z};X\cdot m)$ to approximate this optimal value. To make $I(\widetilde{X \cdot (1-m)};X\cdot(1-m))$ more accurate, it is crucial to accurately estimate $I(\widetilde{X\cdot(1-m)};X\cdot m))$.
> >
> > Following our revised proof in the paper, we derive that mitigating the bias $r$ is essential for improving this estimation. Specifically,
> > * In the first stage (Appendix A1.1), we demonstrate the relationship between optimizing MAE (i.e., minimizing the Lagrangian term) and $I(\widetilde{X\cdot(1-m)};X\cdot m))$.
> > * In the second stage (Appendix A1.3), we show how $I(\widetilde{X\cdot(1-m)};X\cdot m))$ is estimated via $\hat{I}(\hat{z};X\cdot m)$, and how mitigating $r$ reduces the bias in this empirical estimation. This directly improves the estimation of the mutual information and, consequently, MAE performance.
> >
> > ---
> >
> > **R3:**
> >
> > > **1. Why $I(\hat{z} ;X\cdot m|r) = I(\zeta; X\cdot m) - I(\hat{z}; X \cdot m)$?**
> >
> > $r$ is defined as the bias between the optimal solution and the empirical estimation. Therefore, the conditional mutual information $I(\hat{z};X \cdot m|r)$ quantifies the bias between the mutual information of the true effective description I(\zeta; X\cdot m) and the empirical estimation $I(\hat{z}; X \cdot m)$.
> >
> > > **2. Assumption about $r$.**
> >
> > The assumption about $r$ is implicitly addressed in Assumption 3. Since $r$ represents the bias between $\hat{z}$ and $\widetilde{X\cdot (1-m)}$, minimizing the reconstruction loss inherently minimizes this bias. Specifically, Assumption 3 ($\mathcal{L}_{rec}\leqslant\epsilon_l$) implies that $r$ is already reduced to the small value.
> >
> > > **3. Why consider a fixed $r$?**
> >
> > In our Assumption 3, we assume that $\mathcal{L}_{rec}$ is minimized. Once $\mathcal{L}$ is minimized, the bias $r$ is implicitly reduced to a negligible value. At this point, $r$ becomes stable and does not vary significantly with changes in the model. Thus $r$ can be treated as fixed to better isolate and examine other objectives for MAE.
> >
> > > **4. Why $I(\hat{z}; X \cdot m) = I(\zeta; X⋅m) \cap I(\zeta, \hat{z})$?**
> >
> > We apologize for the confusion caused by the previous description. To clarify, this statement was incorrect. The correct formulation is:
> >
> > $I(\hat{z};X \cdot m) = I(\\zeta \cdot m) - (H(\zeta) - I(\zeta ; \hat{z}))$.
> >
> > Therefore, increasing $I(\zeta; \hat{z})$ helps increase $I(\hat{z};X \cdot m)$.

---

> > ### Author Response · Authors · 2024-11-26
> > **Follow-Up Response (Part 2)**
> >
> > > **R5: Could the authors point out any results from Shamir et al. 2010 that directly show $I(\hat{X}; Y) \le \hat{I}(\hat{X}; Y) + O(\frac{K|y|}{\sqrt{n}})$?**
> >
> > Thank you for pointing out this issue. We apologize for the incorrect citation in the paper. The correct derivation follows the intermediate result from Tishby & Zaslavsky (2015) [1]. In their work, they showed that:
> >
> > $I(\hat{X};Y) \leqslant \hat{I}(\hat{X};Y))+O(\frac{K|\mathcal{Y}|}{\sqrt{n}})$.
> >
> > In our paper, we use the form $I(\hat{X}; Y) \leqslant \hat{I}(\hat{X}; Y) + O(\frac{K|y|}{\sqrt{n}}$) in Appendix A.1.2, which directly follows their conclusion. We have updated the citation in the revision to reflect this correction.
> >
> > This formulation is widely used in mutual information estimation papers, including MINE [2] and other subsequent works that build upon it.
> >
> > **References**
> >
> > [1] Naftali Tishby and Noga Zaslavsky. Deep learning and the information bottleneck principle. In 2015 IEEE Information Theory Workshop (ITW), pp. 1–5, 2015. doi: 10.1109/ITW.2015.7133169.
> >
> > [2] Mohamed Ishmael Belghazi, Aristide Baratin, Sai Rajeshwar, Sherjil Ozair, Yoshua Bengio, Aaron Courville, and Devon Hjelm. Mutual information neural estimation. In International conference on machine learning, pp. 531–540. PMLR, 2018.
> >
> > ---
> >
> > **Your timely and insightful feedback has been invaluable in helping us improve our equations and refine our analysis. We sincerely appreciate your support and guidance. If you have any further questions or suggestions, please do not hesitate to let us know.**
> >
> > Best regards,
> > Authors

---

> > > ### Comment · Reviewer_MLQd · 2024-11-28
> > > **Thank you**
> > >
> > > The reviewer truly appreciates the tremendous effort the authors put into clarifying the questions. The revision makes the paper much stronger. The reviewer recommends an acceptance with a score of 8.

---

### Official Review · Reviewer_w8mj · 2024-11-04

**Soundness:** 3
**Presentation:** 3
**Contribution:** 3
**Rating:** 6
**Confidence:** 3

**Summary:**

This work proposes a theoretical analysis of masked autoencoders (MAEs) based on information bottleneck theory. Based on the analysis, it indicates that MAE requires to balance the influences of relevant information and irrelevant information when optimizing the latent space. Therefore, the authors propose two loss functions to satisfy the information bottleneck constraints (One is to retain maximize relevant information between them and the output, and the other is to minimize irrelevant information between them and the input). Experiments show that the method outperforms MAE on image classification, object detection, and semantic segmentation

**Strengths:**

1.	This work proposes a theoretical analysis on the performance of MAE. It has proven that MAE under information bottleneck theory can achieve better performance theoretically.
2.	A novel but simple architecture is proposed to apply information bottleneck theory to MAE, which can improve its performance.
3.	Experiments are conducted on diverse tasks, which is convinced to prove the claim.

**Weaknesses:**

1.	The illustration of the model in Section 4.2 is not very clear. I am not sure about why the architecture can achieve the separation of the relevant and irrelevant parts of the latent space.
2.	There are several studies that introduce the isolation of the latent space with VAE, GAN, or diffusion model. Therefore, I am not sure about the novelty of the proposed MI-MAE in visual tasks.
3.	Experiments only cover several general visual tasks and the results seem not to be significantly better than MAE and other baselines.

**Questions:**

1.	Can you further illustrate how two new losses (l_max_mi, l_min_mi) work to constrain the hidden space that can separate relevant and irrelevant variables?
2.	What is the difference of the work compared with VAE, GAN, or diffusion model that can constrain latent space as well?
3.	The performance improvement of MI-MAE is not significant enough. I guess the assumptions of this work are too idealized, which is not a better method for application. Besides, although the theoretical analysis has revealed the relationship of MAE and information bottleneck theory, the solution is similar to other methods with latent constraints. Can you provide more experimental evidence to show the effectiveness of MI-MAE and the difference of the model with other methods with latent constraints, such as case studies, comparisons with more baselines.

---

> ### Author Response · Authors · 2024-11-21
> **Authors' Response to Reviewer w8mj (Part 1)**
>
> Dear Reviewer w8mj,
>
> We sincerely thank you for your insightful comments and constructive feedback. Below, we address your comments and questions in detail.
>
> > **Q1: Why can the architecture achieve the separation of the relevant and irrelevant parts of the latent space?**
>
> **R1:** Thank you for your question. The separation of relevant and irrelevant parts in the latent space is achieved through the combination of the information bottleneck (IB) principle and the encoder-decoder design. According to the IB principle, relevant information is the part of the image that impacts predictions, while irrelevant information can be discarded without affecting the output. Our method explicitly minimizes the mutual information between the input and the latent representation, which reduces redundant or irrelevant details, and simultaneously maximizes the mutual information between the latent representation and the output, ensuring that only meaningful information is preserved. The encoder compresses the input, guided by these objectives, while the decoder reconstructs the output by relying solely on the retained relevant information. This design naturally enforces the separation of relevant and irrelevant parts of the latent space, as supported by our ablation studies and empirical results, which show improved performance and accurate reconstruction of meaningful features.
>
> ---
>
> > **Q2: Novelty of MI-MAE compared to other studies that introduce the isolation of the latent space with VAE, GAN, or diffusion model.**
>
> **R2:** Thank you for highlighting this point. To the best of our knowledge, MI-MAE is the first work to comprehensively explain the mechanisms of MAE through the lens of the information bottleneck principle. This theoretical foundation sets it apart from previous approaches.
>
> Building on this understanding, we introduce two novel objectives in MI-MAE: (1) minimizing the mutual information between the input and the latent space, and (2) maximizing the mutual information between the latent space and the output. These dual objectives differentiate MI-MAE from prior mutual information-based methods, such as VMI-VAE [1], InfoGAN [2], and InfoDiffusion [3]. Specifically, these earlier works focus solely on maximizing the mutual information between latent representations and outputs (observations), without addressing the minimization of input-latent mutual information or the trade-off described by the information bottleneck principle.
>
> ---
>
> > **Q3: Experiments only cover several general visual tasks and the results seem not to be significantly better than MAE and other baselines.**
>
> **R3:**
> 1. _Scope of visual tasks:_ We closely follow the standard evaluation protocols established in prior MAE studies, including MAE (2022), SimMIM (2022), C-MAE (2023), MFF (2023), and PixMIM (2024). Specifically, by conducting all the tasks (ImageNet linear probing and fine-tuning, COCO detection, ADE20K segmentation) appear in these methods, we believe that our results are directly comparable and sufficiently validate the superiority of MI-MAE.
> 2. _Improvements over baselines:_ On ImageNet fine-tuning, our method achieves a notable improvement of 0.6% over MAE. While this might seem modest, it is double the improvement reported by MFF (CVPR 2023), which achieved only a 0.3% gain. Furthermore, on tasks where pre-training plays a more critical role, such as linear probing and 1% fine-tuning, MI-MAE demonstrates substantial gains, with over 2% improvement on 800-epoch pre-training. These results clearly highlight the effectiveness and superiority of our method compared to previous approaches.
>
> ---
>
> > **Q4: The assumptions of this work are too idealized.**
>
> **R4:** The key assumption in our work, referred to as Assumption 3 in Section 4.3, describes a network within the hypothesis class that minimizes the empirical loss. Under this assumption, the reconstruction loss is constrained to a small value.
>
> This assumption is not overly idealized but aligns with real-world observations. For example, in Figure 3 of the Appendix, we report the reconstruction loss on the ImageNet dataset. The observed reconstruction loss is consistently below the threshold specified by our assumption, $\epsilon_l$. This empirical evidence demonstrates that the assumption is satisfied in practice and accurately reflects the behavior of the network during training. We hope this clarifies that our assumption is both realistic and supported by experimental results.

---

> ### Author Response · Authors · 2024-11-21
> **Authors' Response to Reviewer w8mj (Part 2)**
>
> > **Q5: Besides, although the theoretical analysis has revealed the relationship of MAE and information bottleneck theory, the solution is similar to other methods with latent constraints.**
>
> **R5:** Thank you for your observation. While our method shares a general focus on latent space constraints, it introduces distinct objectives that differentiate it from other latent constraint methods, such as those used in GANs, VAEs, and diffusion models. Specifically, MI-MAE optimizes both mutual information maximization and minimization, addressing a dual objective that is not explored in these other methods (see R2 for detailed comparisons).
>
> In the context of MAEs, previous works have primarily reformulated them through contrastive learning frameworks. In contrast, our work provides a more foundational explanation of MAE mechanisms using the information bottleneck principle. This theoretical perspective enables us to derive two novel objectives: minimizing input-latent mutual information and maximizing output-latent mutual information. They are unique to our approach and distinct from existing methods.
>
> These differences underscore the originality and theoretical depth of our work compared to previous methods involving latent constraints.
>
> ---
>
> > **Q6: More experimental evidence to show the effectiveness of MI-MAE and difference with other methods with latent constraints.**
>
> **R6:**
> 1. _Comparisons with MAE baselines._ We compare the performance of MI-MAE on ImageNet with existing MAE methods that incorporate latent constraints, including C-MAE [4], U-MAE [5], and LC-MAE [6]. The results demonstrating our superiority are summarized below:
> |Method|Epochs|FT|LN|
> |:--:|:--:|:--:|:--:|
> |MAE|100|82.9|55.4|
> |C-MAE|100|82.9|41.1|
> |U-MAE|100|83.0|58.5|
> |LC-MAE|100|83.0|-|
> |MI-MAE|100*|**83.4**|**59.0**|
>
>     *: We used 256 images per batch and reduced the pre-training to 25 epochs to ensure the number of iterations and training samples are equivalent to other methods.
>
> 2. _Comparisons with mutual information maximization._ In our ablation study (Table 3), we evaluated a variant of MI-MAE with only the mutual information maximization (max_mi) loss. This configuration mirrors methods like InfoGAN that focus solely on maximizing mutual information. This variant achieved 82.5% accuracy, while our full method, incorporating both max_mi and min_mi losses, achieved 82.8%. The additional improvements validate the benefits of our dual-objective approach over single-objective methods.
>
> These experiments substantiate the effectiveness of MI-MAE and highlight its advantages over existing methods with latent constraints. If you have specific methods you would like us to include in our comparisons, we would be happy to address them.
>
>
> **References**
>
> [1] Serdega, A. and Kim, D.S., 2020. VMI-VAE: Variational mutual information maximization framework for vae with discrete and continuous priors. arXiv preprint arXiv:2005.13953.
>
> [2] Chen, X., Duan, Y., Houthooft, R., Schulman, J., Sutskever, I. and Abbeel, P., 2016. Infogan: Interpretable representation learning by information maximizing generative adversarial nets. Advances in neural information processing systems, 29.
>
> [3] Wang, Y., Schiff, Y., Gokaslan, A., Pan, W., Wang, F., De Sa, C. and Kuleshov, V., 2023, July. Infodiffusion: Representation learning using information maximizing diffusion models. In International Conference on Machine Learning (pp. 36336-36354). PMLR.
>
> [4] Xiangwen Kong and Xiangyu Zhang. Understanding masked image modeling via learning occlusion invariant feature. In Proceedings of the IEEE/CVF Conference on Computer Vision and Pattern Recognition, pp. 6241–6251, 2023.
>
> [5] Zhang, Q., Wang, Y. and Wang, Y., 2022. How mask matters: Towards theoretical understandings of masked autoencoders. Advances in Neural Information Processing Systems, 35, pp.27127-27139.
>
> [6] Yue, X., Bai, L., Wei, M., Pang, J., Liu, X., Zhou, L. and Ouyang, W., 2023. Understanding Masked Autoencoders From a Local Contrastive Perspective. arXiv preprint arXiv:2310.01994.

---

> ### Author Response · Authors · 2024-11-28
> **Discussion**
>
> Dear Reviewer w8mj,
>
> We sincerely thank you for your efforts in reviewing our paper. We have provided corresponding responses and results, which we believe have covered your concerns. We hope to further discuss with you whether your concerns have been addresses or not. Your feedback is invaluable for us to improve our paper.
>
> Please let us know if you still have any unclear part of our work.
>
> Best,
> Authors

---

> ### Comment · Area_Chair_djjq · 2024-11-28
> **Reviewer feedback and discussion**
>
> Dear Reviewer w8mj,
>
> As the discussion period will end next week, please take some time to read the authors' rebuttal and provide feedback as soon as possible. Did the author address your concerns, and do you have further questions?
>
> Thanks,
>
> Area Chair

---

### Author Response · Authors · 2024-11-21
**Summary of Revisions and Updates**

Dear reviewers,

We sincerely appreciate your insightful and constructive feedback, which has greatly helped us refine our paper in terms of clarity, writing quality, and experimental evaluations. We have carefully addressed your comments in our individual responses to each of your posts. Additionally, we have revised the paper and highlighted the changes in blue fonts. Below is a summary of the updates:
1. We added detailed explanations and mathematical derivations related to our theorems.
2. We unified the notations in Eq. (11) for the overall loss and clearly distinguished batch-level losses ($\mathcal{L}$) from sample-level losses ($l$).
3. We included robustness evaluations on ImageNet-A and ImageNet-C, demonstrating that MI-MAE significantly improves the robustness of MAE across these benchmarks.

Thank you once again for your valuable feedback and suggestions, which have significantly improved the quality of our work.

Best regards,
The Authors

---

> ### Author Response · Authors · 2024-11-26
> **Summary of Changes in the 2nd Revision**
>
> We provide the second revision of our manucript, which covers the issues pointed out by Reviewer MLQd. We sincerely thank all the reviewers again for their valuable feedback for improving the quality of our paper.
>
> The key changes in this new revision are summarized as follows:
> * **Bias in Mutual Information Estimation:** Added a proof in Appendix A.1.2 demonstrating the existence of the bias.
> * **Proof of How Equation 4 Influence the Langrandian Term:** Added a proof in Appendix A.1.1 for better understanding of the influence of reducing bias $r$.
> * **Clarification of Theorem 2:** Revised the proof to explicitly include missing steps, explaining how the bias term affects mutual information and refining the derivations in Eq. (18).
> * **Corrected Citation:** Replaced the incorrect citation of Shamir et al. (2010) with Tishby & Zaslavsky (2015) and updated the references accordingly.
>
> Best regards,
> The Authors

---

### Meta-Review · Area_Chair_djjq · 2024-12-19

**Metareview:**

This paper proposes MI-MAE, a masked image modeling method that learns mask invariant mutual information based on information bottleneck theory. All reviewers give positive scores, recognizing its originality and efficacy. The questions about theoretical soundness, experimental evaluations, and other concerns are mostly addressed in the rebuttal. This paper receives four positive reviews. Therefore, the area chair would recommend accepting this paper.

**Additional Comments On Reviewer Discussion:**

Reviewers proposed various concerns on theoretical soundness, experimental validations, and other aspects. The authors addressed those concerns during rebuttal. Three reviewers acknowledged that their concerns have been solved after the rebuttal and either maintain a positive score or increase their scores. Unfortunately, reviewer w8mj, despite being reminded by authors and area chair for multiple times, failed to respond during the rebuttal and discussion period. The area chair has checked his questions and the rebuttal by authors and confirmed that there's no significant issues remained.

---

### Decision · Program_Chairs · 2025-01-22

Accept (Poster)